# Contribution of solitons to enhanced rogue wave occurrence in shallow depths: a case study in the southern North Sea

**Ina Teutsch**[1]**, Markus Brühl**[2]**, Ralf Weisse**[1]**, and Sander Wahls**[2]

[1]Coastal Climate and Regional Sea Level Changes, Helmholtz-Zentrum Hereon,
Max-Planck-Str. 1, 21502 Geesthacht, Germany
[2]Delft Center for Systems and Control, Delft University of Technology, 2628 CD Delft, South Holland, the Netherlands

**Correspondence:** Ina Teutsch (ina.teutsch@hereon.de)

**Abstract.** The shallow waters off the coast of Norderney in the southern North Sea are characterised by a higher frequency of rogue wave occurrences than expected. Here, rogue waves refer to waves exceeding twice the significant wave height. The role of nonlinear processes in the generation of rogue waves at this location is currently unclear. Within the framework of the Korteweg–de Vries (KdV) equation, we investigated the discrete soliton spectra of measured time series at Norderney to determine differences between time series with and without rogue waves. For this purpose, we applied a nonlinear Fourier transform (NLFT) based on the Korteweg–de Vries equation with vanishing boundary conditions (vKdV-NLFT). At measurement sites where the propagation of waves can be described by the KdV equation, the solitons in the discrete nonlinear vKdV-NLFT spectrum correspond to physical solitons. We do not know whether this is the case at the considered measurement site. In this paper, we use the nonlinear spectrum to classify rogue wave and non-rogue wave time series. More specifically, we investigate if the discrete nonlinear spectra of measured time series with visible rogue waves differ from those without rogue waves. Whether or not the discrete part of the non-linear spectrum corresponds to solitons with respect to the conditions at the measurement site is not relevant in this case, as we are not concerned with how these spectra change during propagation CEI. For each time series containing a rogue wave, we were able to identify at least one soliton in the nonlinear spectrum that contributed to the occurrence of the rogue wave in that time series. The amplitudes of these solitons were found to be smaller than the crest height of the corresponding rogue wave, and interaction with the continuous wave spectrum is needed to fully explain the observed rogue wave. Time series with and without rogue waves showed different characteristic soliton spectra. In most of the spectra calculated from rogue wave time series, most of the solitons clustered around similar heights, but the largest soliton was outstanding, with an amplitude significantly larger than all other solitons. The presence of a clearly outstanding soliton in the spectrum was found to be an indicator pointing towards the enhanced probability of the occurrence of a rogue wave in the time series. Similarly, when the discrete spectrum appears as a cluster of solitons without the presence of a clearly outstanding soliton, the presence of a rogue wave in the observed time series is unlikely. These results suggest that soliton-like and nonlinear processes substantially contribute to the enhanced occurrence of rogue waves off Norderney.

## 1 Introduction

Rogue waves are commonly defined as individual waves exceeding twice the significant wave height, where the significant wave height refers to the average height of the highest one-third of waves in a record. The occurrence of a rogue wave is a rare incident in the framework of a second-order process (Haver and Andersen, 2000). However, due to their exceptional height and unexpected nature, they pose a threat to ships and offshore platforms (Bitner-Gregersen and Gramstad, 2016). Rogue waves have not only been observed in the deep and shallow water depths of the ocean but also approaching coastlines (Didenkulova, 2020). There has been

a lively discussion regarding whether the occurrence frequency of rogue waves in the open ocean is well described by common wave height distributions. Both Rayleigh (Longuet-Higgins, 1952) and Weibull distributions (Forristall, 1978), which are based on the linear superposition of wave components, have been used to describe the distributions of wave and crest heights. Later theories include second-order steepness contributions in wave height distributions (e.g. Tayfun and Fedele, 2007). Distributions have been assessed for measurement data collected by surface-following buoys (e.g. Baschek and Imai, 2011; Pinho et al., 2004; Cattrell et al., 2018), radar devices (e.g. Olagnon and van Iseghem, 2000; Christou and Ewans, 2014; Karmpadakis et al., 2020), laser altimeters (e.g. Soares et al., 2003; Stansell, 2004), and acoustic Doppler current profilers (ADCPs) (Fedele et al., 2019). Independent of the measurement device, some authors have found measured wave heights to agree well with the established distributions (e.g. Casas-Prat et al., 2009; Waseda et al., 2011; Christou and Ewans, 2014), whereas others have found the frequency of rogue wave occurrences to be overestimated (e.g. Olagnon and van Iseghem, 2000; Baschek and Imai, 2011; Orzech and Wang, 2020) or underestimated (e.g. Stansell, 2004; Pinho et al., 2004). Numerous authors have described local differences in the rogue wave occurrence frequency between their measurement stations (Baschek and Imai, 2011), depending on the wave climate (Stansell, 2004), especially in coastal waters, where waves interact with the seabed (Cattrell et al., 2018; Orzech and Wang, 2020). Massel (2017) stated that the wave height distribution is dependent on the water depth; however, the water depth is not explicitly included in the common models. Karmpadakis et al. (2020) found that, while different models can describe wave height distributions well within narrow ranges of sea state conditions, no model is able to describe measured wave heights for a wide range of sea states accurately. Mendes and Scotti (2021) recently introduced a new exceedance probability distribution for rogue waves by geometrically combining some commonly used distributions. This combined distribution is more flexible than the individual distributions, as it is additionally dependent on sea state variables. The distribution is capable of describing rogue waves in a wide range of sea states and was also able to describe the uneven rogue wave distributions in storms that were observed by Stansell (2004).

In a previous study, we analysed measurement data from various stations in the southern North Sea (Teutsch et al., 2020) and found the rogue wave occurrence frequencies to vary spatially and by measurement device. For data obtained from wave buoy measurements, we generally found rogue wave frequencies to be slightly overestimated by the Forristall distribution, which is a special form of the Weibull distribution, fit to wave data recorded during hurricanes (Forristall, 1978). An exception was one measurement buoy that was located in the shallow waters off the coast of the island of Norderney, Germany (Fig. 1). For this buoy, enhanced

rogue wave occurrence was observed that could not be explained by the Forristall distribution. This suggests that nonlinear processes and interactions may play a role in increasing the rogue wave occurrence frequency at this specific location. In order to better understand the impact of nonlinear processes at this location, we analyse surface elevation time series from this location using a so-called nonlinear Fourier transform[1] (NLFT) (Ablowitz et al., 1974; Osborne, 2010). Different NLFTs exist for different wave evolution equations and boundary conditions. Therefore, before the contributions of our work are detailed, we first discuss the most common NLFTs and their use in connection with rogue waves.

To date, the nonlinear behaviour of deep-water rogue waves has received considerably more attention than that of shallow-water rogue waves. The evolution of the complex envelope of unidirectional wave trains in deep water can be described by the cubic nonlinear Schrödinger (NLS) equation (Zakharov, 1968; Whitham, 1974). The NLS equation is a weakly nonlinear, narrow-banded approximation of the fully nonlinear water wave equations (Whitham, 1974) that can be solved exactly using an appropriate NLFT (Zakharov and Shabat, 1972). In deep water, rogue wave occurrence beyond the second-order model has been explained, for example, by a nonlinear instability that was also found in numerical simulations and tank experiments (see e.g. Dysthe et al., 2008, and the references therein). Here, uniform wave trains exhibit modulational instability with respect to small side-band perturbations and disintegrate into groups, in which the highest wave becomes significantly larger than the wave height in the original train (Benjamin and Feir, 1967). The instability is, therefore, also known as modulational instability. The NLS equation has exact solutions – known as breathers – that have been suggested as an analytical model of rogue waves in a unidirectional case (Dysthe and Trulsen, 1999). Just like rogue waves, breathers seem to "appear from nowhere and disappear without a trace" (Akhmediev et al., 2009). This impressive effect was demonstrated experimentally by Chabchoub et al. (2011). Slunyaev and Shrira (2013) investigated the behaviour of breathers beyond the NLS equation numerically, using the full two-dimensional Euler equations. Breather solutions are known to occur after the modulational instability has been triggered for randomly perturbed plane waves (e.g. Soto-Crespo et al., 2016; Randoux et al., 2016; Grinevich and Santini, 2018). Furthermore, it was found that random sea states can lead to similar results (Onorato et al., 2001, 2006). However, the relevance of modulational instability for the formation of oceanic rogue waves has been doubted based on the analysis of real-world events (Fedele et al., 2016). A recent review in Dudley et al. (2019) discusses these and many other works in this area.

The form of the NLFT for the NLS equation (NLS-NLFT) depends on the boundary conditions. Initially, the NLS-

_______________

[1]Nonlinear Fourier transforms are also known as scattering transforms in the literature.

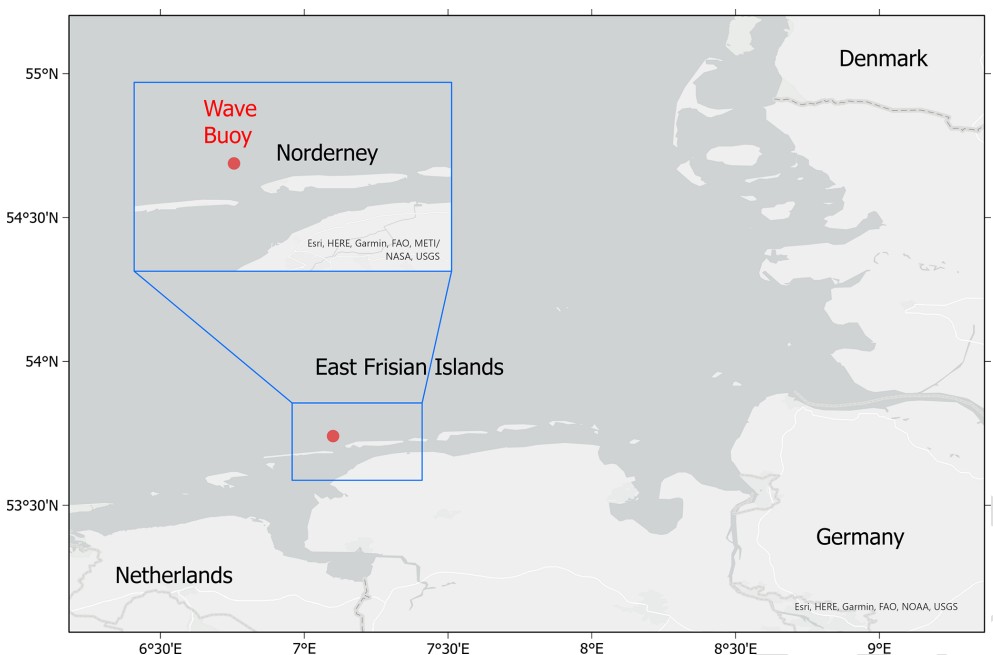

**Figure 1.** Map of the German Bight, showing the location of the measurement buoy close to the island of Norderney.

NLFT was developed for vanishing boundary conditions, where localised wave packets with sufficient decay are considered (Zakharov and Shabat, 1972). The NLS-NLFT for vanishing boundary conditions decomposes a wave packet into solitons and a radiative part (Ablowitz and Segur, 1981). An NLS-NLFT for periodic boundary conditions was developed by Its and Kotlyarov (1976) (see Kotlyarov and Its, 2014, for an English translation). The periodic NLS-NLFT instead represents a periodic wave using Riemann theta functions. This representation can be interpreted as nonlinearly interacting stable modes (i.e. Stokes waves) and unstable modes (Osborne, 2010). Special solutions of the NLS equation such as solitons and breathers have distinctive representations in both nonlinear Fourier domains (Osborne, 2010). Therefore, the periodic NLS-NLFT has been used to analyse rogue wave data by various authors. Osborne et al. (2000) proposed the interpretation of unstable modes in the nonlinear Fourier spectrum as (potentially small) rogue wave components (also see Osborne, 2010). A recent study of a real storm using this approach was presented in Osborne et al. (2019). With the help of the periodic NLS-NLFT, Islas and Schober (2005) observed that rogue waves in random Joint North Sea Wave Project (JONSWAP) data are close to homoclinic solutions of the NLS equation (also see Calini and Schober, 2017). Randoux et al. (2016) proposed classifying rogue waves based on the periodic NLS-NLFT of their local periodisation and applied this technique to rogue waves formed in simulations of a dam break and the modulational instability. In Randoux et al. (2018), this technique was applied to experimental data of Peregine breathers. Onorato

et al. (2021) applied it to a giant wave packet measured in the ocean.

The vanishing NLS-NLFT, which detects envelope solitons and radiation in deep-water wave packets, has been applied to rogue waves as well. As pointed out by Slunyaev (2006), the vanishing NLS-NLFT is easier to compute and interpret. Furthermore, breather solutions typically consist of one or more solitons that interact with a periodic background (Slunyaev, 2006). In Slunyaev (2006), the NLS-NLFT was used to detect envelope solitons for a measured rogue wave and estimate their parameters (e.g. amplitude, velocity and position). Slunyaev (2018) estimated the accuracy of this procedure for strongly nonlinear waves. The NLFT was applied to the interpretation of deep-water waves, the extraction of soliton-like groups and the prediction of their further dynamics. Carrying this work further, Slunyaev (2021) identified a wave group in numerical simulations as a stable envelope soliton, which could be related to rogue wave events. In addition to the periodic NLS-NLFT, Onorato et al. (2021) also applied the vanishing NLS-NLFT to the giant wave packet.

The role of nonlinear processes with respect to rogue wave generation in shallow water has received considerably less attention than for deep water. Shallow-water wind waves substantially differ from deep-water wind waves; therefore, it is not appropriate to simply scale the deep-water nonlinear interaction to shallow-water waves (Janssen and Onorato, 2007). As the water depth becomes more and more shallow, a wave-induced current develops and less wave energy is available for nonlinear focusing (Benjamin and Feir, 1967; Janssen and Onorato, 2007). Although waves in shallow wa-

ter can also destabilise due to oblique perturbations (Toffoli et al., 2013), the modulational instability in shallow water does not enhance the formation of extreme waves (Fernandez et al., 2014). Didenkulova et al. (2013), supported by observations, reported that the influence of the modulational instability on rogue wave generation becomes less probable in shallow water. Fedele et al. (2019) stated that waves in shallow water break before they can start to "breathe" and become rogue waves. Glukhovskiy (1966) hypothesised early that high individual waves in shallow water would occur less frequently than predicted by the Rayleigh distribution due to depth-induced wave breaking. Therefore, some authors expect the rogue wave probability to decrease in shallow water (e.g. Slunyaev et al., 2016). Other authors have referred to the large ratio between nonlinearity and dispersion in shallow water (Kharif and Pelinovsky, 2003) and have concluded that Gaussian statistics are not sufficient for the description of shallow-water waves and that rogue waves are likely to occur more frequently as the water depth decreases (Garett and Gemmrich, 2009; Sergeeva et al., 2011). The nonlinear processes in shallow water are mainly a result of the interaction of waves with the seafloor (Prevosto, 1998). Refraction, shoaling and higher-order nonlinear effects change the shapes of waves and their energy spectrum (Bitner, 1980; Tayfun, 2008). Soomere (2010) found that additional processes associated with the generation of extreme waves, like wave amplification along certain coastal profiles, redirection of waves or the formation of crossing seas, are more relevant in shallow water (compared with deep water) due to wave–bathymetry interactions; therefore, more rogue waves should be expected in nearshore regions.

In shallow water, the wave evolution is described by the Korteweg–de Vries (KdV) equation (Korteweg and de Vries, 1895). It describes weakly nonlinear and dispersive progressive unidirectional free-surface waves in shallow water with constant depth (Whitham, 1974). Osborne and Petti (1994) point out that $kh$, where $k$ and $h$ represent the wave number and water depth, respectively, should not be much larger than one for the KdV equation because of how the dispersion relation is approximated. The threshold $kh \leq 1.36$ marks the point at which the modulational instability disappears (Osborne and Petti, 1994). Following Osborne (1995), we use this threshold to define shallow-water conditions in this work. The regular wave solutions of the KdV are stable, i.e. the wave amplitude does not change significantly when the initial wave train is perturbed. Therefore, the modulational instability cannot contribute to the explanation of rogue wave occurrence in shallow water.

The KdV equation can again be solved using suitable NLFTs. The NLFT for the KdV equation (KdV-NLFT) with vanishing boundaries was found by Gardner et al. (1967). Its and Matveev (1975) presented the Riemann theta form of the periodic KdV-NLFT. As in the NLS case, the vanishing KdV-NLFT decomposes a signal into solitons and radiation, while the periodic KdV-NLFT can be interpreted as a super-

position of cnoidal waves plus their nonlinear interactions (Osborne, 2010). While there seems to be no work on applying the KdV-NLFT to rogue waves, it has been exploited to investigate potentially hidden solitons in shallow water.

By numerically solving the KdV equation, Zabusky and Kruskal (1965) discussed the decomposition of a cosine signal into a train of eight solitons. They documented that the amplitude and shape of solitons remain unaffected by nonlinear interactions with each other. Osborne and Bergamasco (1986) applied the periodic KdV-NLFT and found it could detect the solitons in the numerical experiment of Zabusky and Kruskal (1965) before they became visible. In Osborne et al. (1991), they used this method to analyse surface-wave data from the Adriatic Sea. Christov (2009) used the periodic KdV-NLFT to analyse internal waves in the Yellow Sea. Costa et al. (2014) used the periodic KdV-NLFT to confirm the soliton content of low-pass-filtered time series measured in the Currituck Sound during a storm. Brühl and Oumeraci (2016) and Trillo et al. (2016) independently confirmed the findings of Osborne and Bergamasco (1986) experimentally. A comprehensive comparison of the vanishing and periodic NLFT with the conventional Fourier transform for the detection of hidden solitons in bores has been presented recently by Brühl et al. (2022).

The nonlinear interaction of solitons in shallow water has been discussed with regard to its role in rogue wave generation. Pelinovsky et al. (2000) showed that dispersive focusing is possible for the vanishing KdV equation, but they also mentioned that "the 'nonlinear' [wave] train should include a soliton". Equivalently to the linear case, in which rogue waves evolve from the superposition of wave components, nonlinear focusing is then the interaction between one or, in principle, multiple solitons with dispersive waves, due to their velocity difference. For the unidirectional case, Kharif and Pelinovsky (2003) found that the interaction of KdV solitons does not lead to a significant increase in surface elevation. Soomere (2010) considered that, as soliton interaction in the unidirectional case does not lead to an enhancement in surface elevation, a higher nonlinearity should even lead to a decrease in the rogue wave occurrence probability. As this is not consistent with observations, the aforementioned author concluded that directionality must play a role in rogue wave generation in shallow water. Indeed, crossing solitons are known to be able to produce large amplitudes (Peterson et al., 2003). In contrast to linear superposition, the interaction of two crossing solitons may produce a crest up to 4 times CE2 higher than the incoming waves (Peterson et al., 2003).

At the moment, rogue wave occurrence in shallow water has not been sufficiently explained beyond second order. Moreover, almost all investigations in previous work have been based on theoretical considerations, numerical simulations or laboratory experiments. In this study, we instead leverage the vanishing KdV-NLFT to analyse the soliton spectrum of a large number of time series with and with-

out rogue waves that have been measured off the coast of Norderney in the southern North Sea. For this location, wave height distributions based on linear superposition have been shown to underestimate rogue wave occurrence (Teutsch et al., 2020). We apply the KdV-NLFT for vanishing boundaries (vKdV-NLFT) as a spectral analysis method to explore the extent to which the presence of solitons might contribute to the enhanced rogue wave occurrence off Norderney. Following Sugavanam et al. (2019), we use the NLFT only as a signal processing tool. Our goal is to classify time series by their nonlinear spectra. We do not assume that the nonlinear soliton spectra remain constant during propagation beyond the measurement site, which would be the case only if the propagation conditions are well approximated by the KdV equation.

The structure of the paper is outlined in the following. Section 2.1 describes the measurement site and the dataset and gives a definition for rogue waves. In Sect. 2.2, the application of vKdV-NLFT to the measurement data is explained. Section 3 consists of two parts: in Sect. 3.1, we explore the direct association of solitons calculated from NLFT with rogue waves, and Sect. 3.2 discusses statistical differences in the soliton spectra of time series with and without rogue waves. In Sect. 4, we discuss the time windows and location for which our results are valid and suggest further investigations. In Sect. 5, our conclusions are presented.

## 2 Methods

### 2.1 Measurement site and dataset

We analysed wave elevation data measured by a surface-following buoy off the coast of the island of Norderney in the German Bight in the time period between 2011 and 2016. The predominant wave propagation direction during this time period was southeast (Fig. 3). The measurement buoy was deployed at a nominal water depth of $h = 10$ m, which was assumed to be constant for the following analyses. In reality, the water depth off the coast of Norderney is not constant, as the bathymetry at the location is spatially highly variable with strong gradients (Fig. 2). The bed slope perpendicular to the wave direction varies between $1 : 500$ (offshore direction) and $1 : 200$ (onshore direction). As the buoy is restricted only by its mooring, there is the possibility that it will move horizontally. The actual water depth $h$ below the horizontally moving buoy may then be subject to rapid changes. In addition, the tidal range at the site is about 2.5 m (NLWKN, 2021), which further causes the water depth to vary.

The wave data were measured at a frequency of 1.28 Hz and are available as a set of time series (samples) of 30 min length. To exclude low-energy sea states in the following, only samples with a significant wave height $H_s$ above the long-term 70th percentile of the significant wave height, $H_{s,70} = 1.29$ m, were included in the analysis. Here, the sig-

nificant wave height $H_s$ is defined as the mean of the highest 30 % of the wave heights in a 30 min sample. $H_{s,70}$ was calculated from the significant wave heights $H_s$ of all 30 min samples during the 6 years of available measurement data. On the one hand, this excludes possible measurement uncertainties caused by short waves that are only described by a few points; on the other hand, it includes only rogue waves of heights relevant for offshore activities. As the KdV equation for shallow water was to be applied to the data, only samples satisfying shallow-water conditions in terms of the validity of the KdV equation were included in the study. The definition of shallow water depths for the applicability of the KdV equation is different from the commonly used definition of shallow water in the engineering context, $kh < \pi/10$ (Dingemans, 1997). As explained in Sect. 1, the shallow-water condition used in this study was

$$\frac{h}{L} < 0.22 \quad \text{or} \quad kh \leq 1.36, \tag{1}$$

with water depth $h$ and wavelength $L$. The wavelength was calculated as

$$L = T_p \cdot c \tag{2}$$

from the peak period $T_p = f_p^{-1}$ of each sample, where $f_p$ is the peak frequency in the linear fast Fourier transform (FFT) spectrum of the sample, and the linear phase speed $c = \sqrt{gh}$, where $g$ is gravity. Following Eqs. (1) and (2), the condition for the peak period may be written as

$$T_p > \frac{h}{0.22 \cdot c}. \tag{3}$$

Thus, for a water depth of $h = 10$ m, the peak period had to satisfy the condition $T_p > 4.6$ s in order for a sample to classify for shallow depth conditions in which the KdV equation is valid. We based the shallow-water condition on the peak period $T_p$ of the entire sample to assume that shallow-water wave properties as described by the KdV equation strongly contribute to the wave processes in the sample. Nevertheless, it was additionally ensured that each of the individual rogue waves (or the highest wave in each sample that did not contain a rogue wave) satisfied the depth conditions required for the applicability of the KdV equation, based on its period $T_{max}$. Of all the selected samples above $H_{s,70}$, the required shallow depth conditions applied in more than 98 % of cases and were, thus, the dominant condition in these samples. The 2 % of the samples not satisfying the condition of shallow depth were discarded and not considered in the analysis. In the considered samples, $kh$ ranged between 0.38 and 1.36.

Rogue waves are commonly defined as waves with an individual height $H$ from crest to trough of (Haver and Andersen, 2000)

$$H \geq 2.0 \, H_s \tag{4}$$

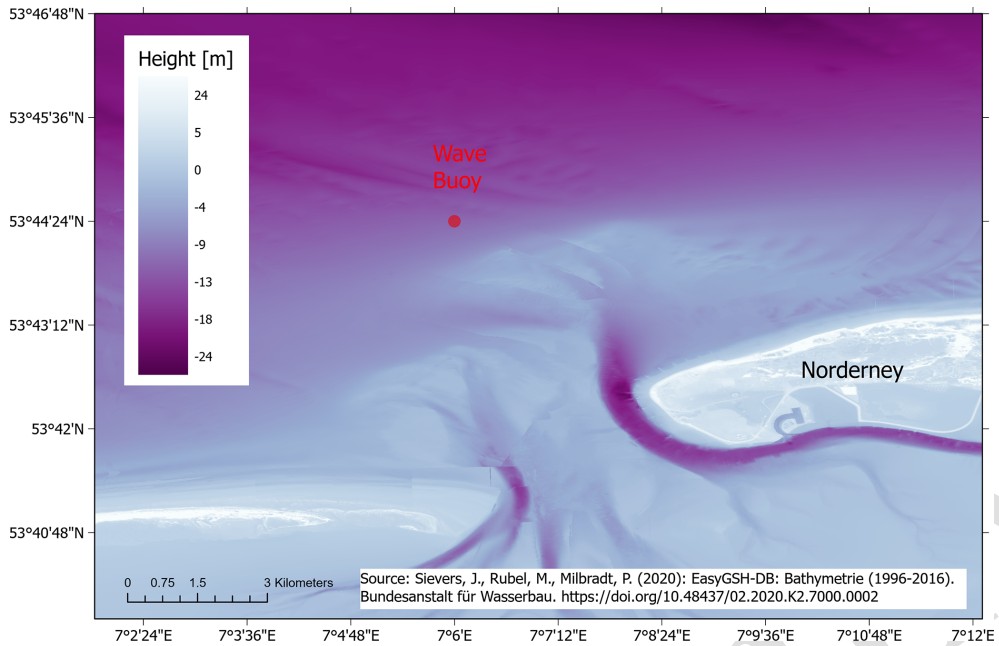

**Figure 2.** Bathymetry conditions at Norderney relative to NHN (Normalhöhennull), which represents the standard elevation zero of the German reference height system, and the position of the measurement buoy.

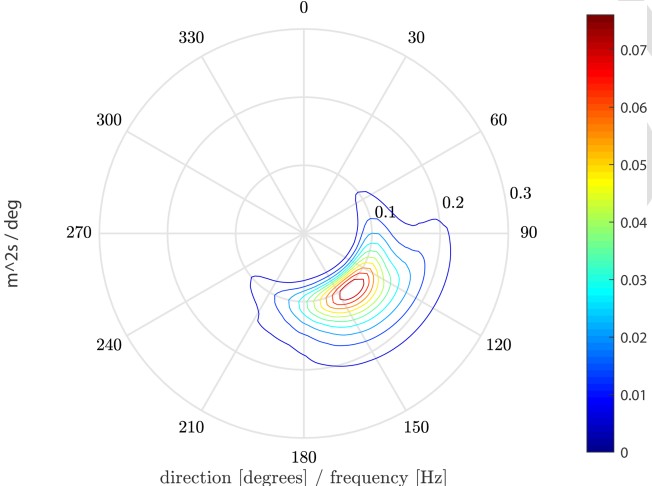

**Figure 3.** Mean directional wave spectrum for the time period from 2011 to 2016, obtained using the DIrectional WAve SPectra Toolbox (DIWASP; Johnson, 2002).

and/or waves with a crest height $C$ above still water level of (Haver and Andersen, 2000)

$$C \geq 1.25\, H_\mathrm{s}. \tag{5}$$

In a previous study based on measurement data from the southern North Sea (Teutsch et al., 2020), we found that the rogue wave frequency significantly deviated from the Forristall distribution for wave heights larger than 2.3 $H_\mathrm{s}$. Therefore, in the present study we further define "extreme rogue

waves" by a more strict height criterion of

$$H \geq 2.3\, H_\mathrm{s}. \tag{6}$$

For the definition of a wave, the zero-upcrossing method was used.

The measured time series were subdivided into five categories:

– *Non-rogue samples* comprise measurement samples that did not include any rogue wave.

– *Height rogue samples* comprise measurement samples that include a rogue wave only according to the height criterion defined in Eq. (4) but exclude the extreme rogue waves according to Eq. (6) and the double rogue samples (see below).

– *Crest rogue samples* comprise measurement samples that include a rogue wave only according to the crest criterion defined in Eq. (5) but exclude the double rogue samples.

– *Double rogue samples* comprise measurement samples that include a rogue wave that fulfilled both the criteria defined in Eqs. (4) and (5) at the same time but exclude the extreme rogue waves according to Eq. (6).

– *Extreme rogue samples* comprise measurement samples that include a rogue wave according to the height criterion defined in Eq. (6) but exclude the double rogue samples.

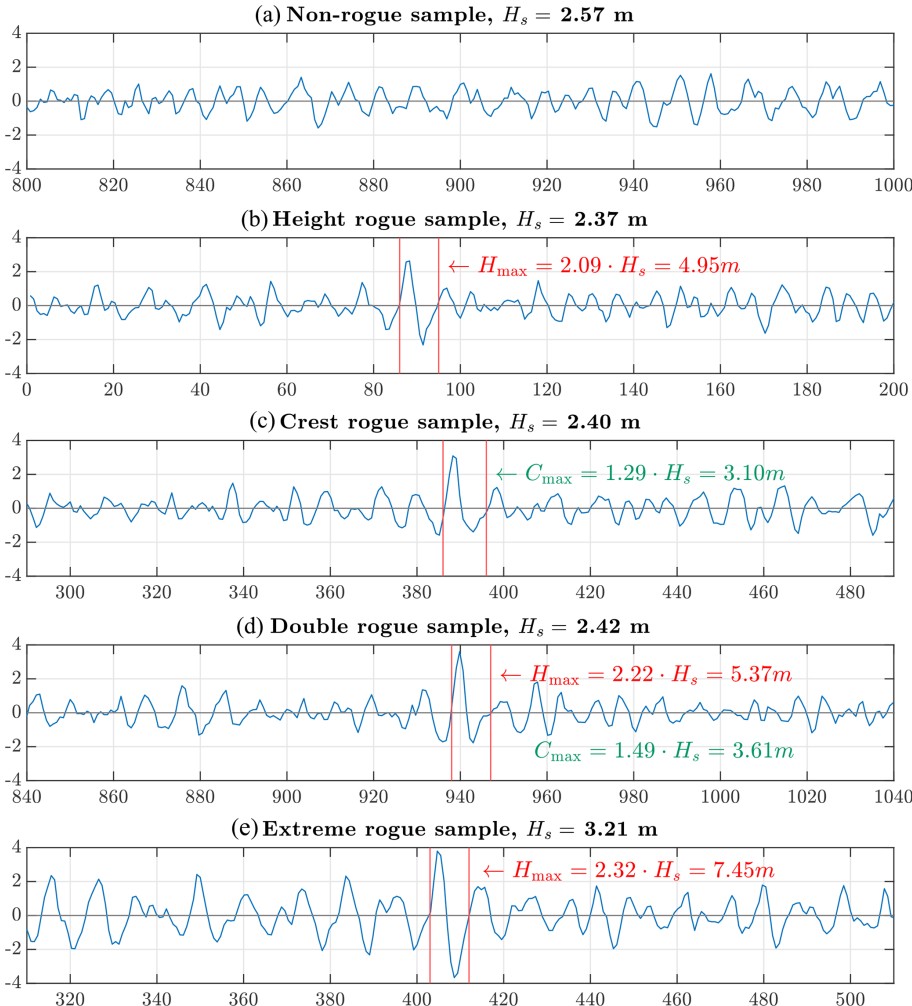

**Figure 4.** Panels **(b)**–**(e)** show 200 s sections taken from example time series illustrating rogue waves for each of the four rogue wave categories, and panel **(a)** presents a non-rogue wave sample with a similar value of $H_s$ for comparison. Vertical red lines mark the two zero-upcrossings of the rogue wave. Rogue wave and crest heights are indicated in red and green, respectively.

**Table 1.** Number of samples and total number of individual waves in the considered time series categories.

| Category | Non-rogue | Height rogue | Crest rogue | Double rogue | Extreme rogue | Total |
|---|---|---|---|---|---|---|
| No. of samples | 13 984 | 833 | 95 | 151 | 93 | 15 156 |
| Total no. of waves | 4 759 663 | 287 617 | 32 354 | 52 520 | 32 117 | 5 164 271 |
| Sample percentage | 92.3 % | 5.5 % | 0.6 % | 1.0 % | 0.6 % | 100 % |

Examples of each time series category are shown in Fig. 4. Table 1 shows the number of samples and the percentage of samples in each category.

## 2.2 Application of the Korteweg–de Vries equation with vanishing boundary conditions to the measurement data

The vKdV-NLFT was applied to the data in order to obtain the discrete soliton spectrum of each time series. The KdV equation was introduced by Korteweg and de Vries (1895). It describes the evolution of weakly nonlinear and dispersive progressive unidirectional free-surface waves in shallow wa-

ter with constant depth. For the analysis of space series (fixed at one point in time), the space-like KdV equation (sKdV) is given e.g. in Osborne (2010) as

$$u_t + c\,u_x + \alpha\,u\,u_x + \beta\,u_{xxx} = 0, \tag{7}$$

in which $u = u(x,t)$ is a free-surface space series, developing in space $x$ and time $t$. The subscripts $x$ and $t$ denote partial derivatives, $c$ is the phase speed in shallow water, and $\alpha = (3c)(2h)^{-1}$ and $\beta = (ch^2)/6$ are constants, depending on the phase speed $c$ and the water depth $h$. Equation (7) can be adapted to the analysis of time series (fixed at one point in space, such as buoy measurements). For the case of a free-surface elevation time series $u(x_0,t)$ (see e.g. Fig. 5) at location $x_0$, the spatial evolution is then described by the time-like KdV equation (tKdV) (Osborne, 1993)

$$u_x + c'\,u_t + \alpha'\,u\,u_t + \beta'\,u_{ttt} = 0, \tag{8}$$

in which $c' = c^{-1} = (\sqrt{gh})^{-1}$, $\alpha' = -\alpha\,(c^2)^{-1}$ and $\beta' = -\beta\,(c^4)^{-1}$. For our application of the KdV-NLFT, we assumed initial conditions with vanishing boundaries, i.e.

$$\lim_{t \to \pm\infty} u(x_0,t) = 0 \tag{9}$$

sufficiently fast. As we were mainly interested in the soliton part of the nonlinear spectrum and solitons are not periodic, we preferred vanishing to periodic boundary conditions. For vanishing boundary conditions, the initial wave packet develops into a train of solitons followed by an oscillatory trail that vanishes over time (e.g. Ablowitz and Segur, 1981). The soliton spectrum therefore completely describes the behaviour of the wave train in the far field. The surface elevation in the far field is then described by

$$u(x,t) \approx \sum_{n=1}^{N} \tilde{u}_n \, \mathrm{sech}^2\left(\omega_n t - k_n x - \phi_n\right), \tag{10}$$

i.e. as the linear superposition of independent solitons after the oscillatory waves have dampened out, with $\tilde{u}_n$ and $k_n$ uniquely determined by $\omega_n$, $h_0$ and $g$ (Ablowitz and Kodama, 1982, Eq. 2.20a). The nonlinear spectrum of the vKdV-NLFT consists of a discrete spectrum representing solitons and a continuous spectrum representing oscillatory waves. We applied the vKdV-NLFT by using the MATLAB (2019) interface to a development version (commit 681191c) of the FNFT software library (Wahls et al., 2018). Figure 5 shows an example of a measured time series, its linear FFT spectrum, the nonlinear continuous spectrum and the discrete nonlinear soliton spectrum. In this paper, only the discrete soliton spectrum will be discussed further. Each of the solitons in the discrete spectrum would be a physical soliton if the signal is propagated according to the KdV equation with vanishing boundary conditions. After sufficiently long propagation, the solitons will separate and their characteristic shapes become clearly visible. For visualisation of the role

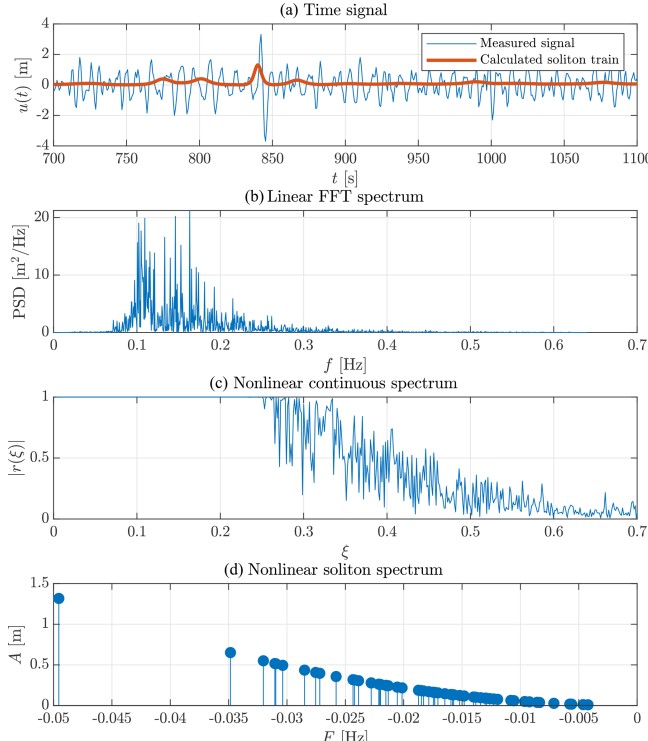

**Figure 5.** Example of a time series including a rogue wave at approximately 820 s, and its corresponding FFT and NLFT spectra. The nonlinear spectra were calculated from vKdV-NLFT. The time series with $H_{\max} H_s^{-1} = 2.58$, $H_{\max} = 7.00\,\mathrm{m}$ and $H_s = 2.71\,\mathrm{m}$ was measured on 17 October 2013, starting at 11:30 CEST. Panel **(a)** additionally shows the soliton train, as obtained by nonlinear superposition of the solitons in the discrete spectrum (Prins and Wahls, 2021). The required soliton phase shifts were computed using the method of Prins and Wahls (2019). Note that inverting large soliton spectra is numerically difficult (Prins and Wahls, 2021); therefore, a shortened time series was used in panel **(a)**.

of solitons in the time series, Fig. 5a shows the soliton train that was obtained by nonlinear superposition of the solitons (considering their interactions but neglecting the continuous spectrum) using the algorithm from Prins and Wahls (2021). Although a soliton does not cross the still water level, a mathematical definition of the angular frequency can be obtained from the soliton solution of the tKdV (Brühl et al., 2022, Eq. 12) as follows:

$$\Omega = 2\pi \cdot F = \sqrt{\frac{3Ag}{4h^2}}. \tag{11}$$

As this equation relates the frequency $F$ to the amplitude $A$ of the soliton, the frequency sorts the solitons in the spectrum by their amplitude. Following the convention in Brühl and Oumeraci (2016), the solitons in the discrete spectrum (Fig. 5d) are displayed on a negative frequency axis. The vKdV-NLFT was applied to all 15 156 samples listed in Table 1.

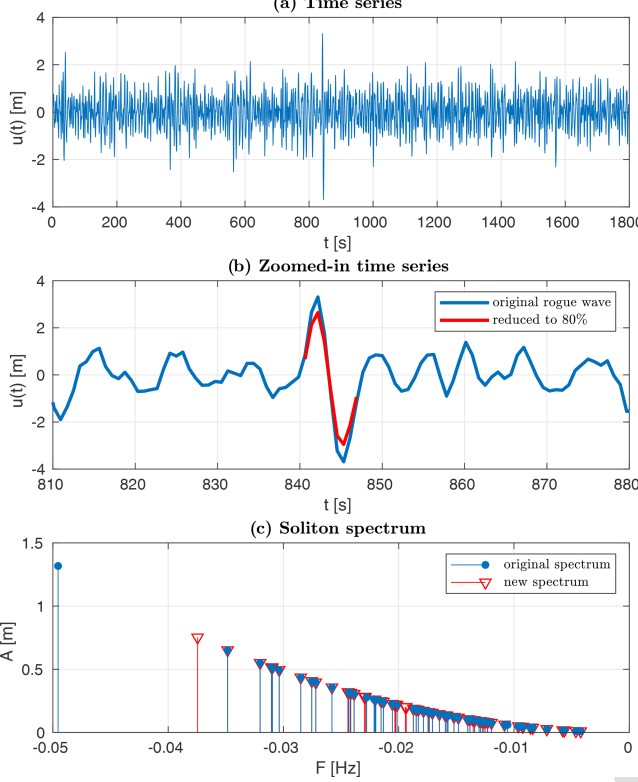

**Figure 6.** Panel **(a)** presents an extreme rogue time series from 17 October 2013, starting at 11:30 CEST. Panel **(b)** displays a magnified view of the rogue wave (blue curve) and the reduction of its elevation to 80 % (red curve). Panel **(c)** shows soliton spectra of the original (blue circles) and the modified time series (red triangles) resulting from vKdV-NLFT.

## 2.3 Attribution of solitons to rogue waves

The aim of the study was to explore the role of the individual solitons in the generation of rogue waves. The following procedure was used to check whether individual solitons in the NLFT spectrum could be associated with the recorded rogue waves. First, the KdV-NLFT of the original time series was computed. Following this, all free-surface elevations between the two zero-upcrossings of a rogue wave (or the largest wave for non-rogue wave samples) were scaled down to 80 % (Fig. 6). The KdV-NLFT was then repeated for the modified time series, which resulted in a new soliton spectrum. We monitored which of the solitons had changed in amplitude $A$ (and, therefore, in frequency $F$), due to the change in wave height of the modified rogue wave. These solitons were assumed to have the same position in the time series as the rogue/maximum wave.

## 3 Results

Regular and irregular wave trains in very shallow water are known to often contain solitons, even without the presence of rogue waves (e.g. Osborne et al., 1991; Brühl and Oumeraci, 2016). Our data support this finding: solitons were found in all samples, with and without rogue waves. In the following, we therefore first investigate whether individual solitons in the NLFT spectrum can be associated with the recorded rogue waves. Afterwards, we explore whether the soliton spectra calculated from rogue wave time series show differences when compared with those calculated from non-rogue wave time series.

### 3.1 Attribution of solitons to rogue waves

Solitons were attributed to specific rogue waves, following the procedure described in Sect. 2.2. In each case, we found that the amplitude of one large soliton significantly decreased for a reduced rogue wave (or maximum wave) height. Furthermore, slight changes in amplitudes were observed in the group of smaller solitons. As amplitude $A$ and frequency $F$ are related according to Eq. (11) for solitons, the reduction in amplitude corresponded to a simultaneous shift in frequency, which can be seen in the soliton spectrum (Fig. 6). The reduced solitons can be regarded as being associated with the rogue wave in the time series, while the other solitons in the spectrum maintained their amplitudes. The solitons with constant amplitudes can be regarded as not being associated with the rogue wave. We refer to the amplitudes of the $l = 1 \dots n$ solitons associated with the rogue wave as $A_S^n$, with $A_S^1$ denoting the largest attributed soliton. Although often the case, the largest soliton attributed to the rogue wave was not necessarily the largest soliton in the spectrum (Fig. 7).

We extracted the amplitude of the largest attributed soliton $A_S^1$ for each time series and compared it to the rogue wave height $H$ (for rogue waves according to any of the two height criteria, including double rogue waves; Fig. 8a) or the crest height $C$ of the rogue wave (for rogue waves according to the crest criterion, including double rogue waves; Fig. 8b). A comparison of the soliton amplitude $A_S^1$ to the largest wave height $H_{\max}$ and the largest crest height $C_{\max}$ in non-rogue wave samples has been added for reference (Fig. 8c, d). The slopes of the linear regression curves express increasing $A_S^1$ with increasing $H$ or $H_{\max}$ and $C$ or $C_{\max}$. For the analysed samples, the scatter of the data suggests an upper limit of $A_S^1$ of between 2 and 3 m. The goodness of fit of each curve to the data is given in terms of the coefficient of determination

$$R^2 = \frac{\text{SS}_{\text{res}}}{\text{SS}_{\text{total}}}, \tag{12}$$

in which $\text{SS}_{\text{res}}$ is the sum of squares of residuals with respect to the regression curve and $\text{SS}_{\text{total}}$ is the sum of squares of residuals with respect to the average value of the data (and thus a measure of the variance). $R^2$ indicates that the linear

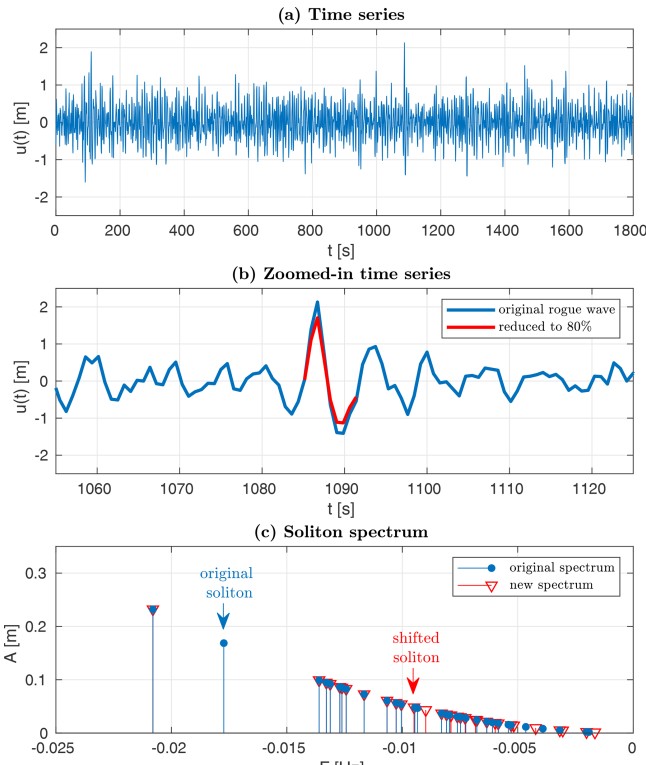

**Figure 7.** Panel **(a)** presents a double rogue time series from 27 April 2016, starting at 20:30 CEST. Panel **(b)** displays a magnified view of the rogue wave (blue curve) and the reduction of its elevation to 80 % (red curve). Panel **(c)** shows soliton spectra of the original (blue circles) and the modified time series (red triangles) resulting from vKdV-NLFT.

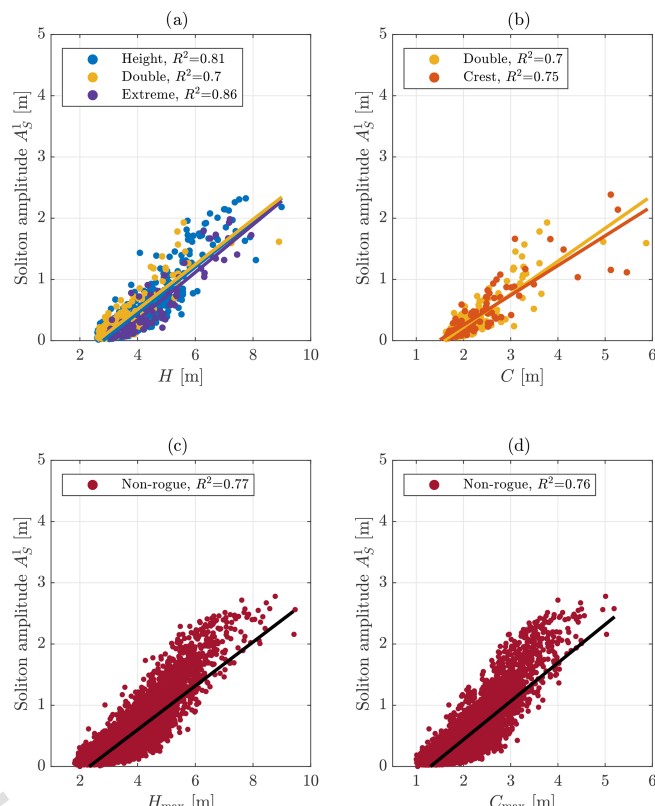

**Figure 8.** Amplitude of the largest soliton attributed to the highest wave, $A_S^1$, in the time series for the rogue wave **(a, b)** or non-rogue wave **(c, d)** samples as a function of rogue wave height $H$ and maximum wave height $H_{\max}$ **(a, c)**, respectively, or rogue crest height $C$ or maximum crest height $C_{\max}$ **(b, d)**, respectively CE3. The goodness of fit of the linear regression curves is given in terms of $R^2$.

curves fit the results from height and extreme rogue samples better than the results from non-rogue, double and crest rogue samples. $R^2$ is higher in Fig. 8a than in Fig. 8b–d.

Moreover, it is seen that the amplitude of the largest soliton is always smaller than the rogue wave crest/height itself. This is in agreement with results by Osborne et al. (1991), who identified solitons in measurement data from the Adriatic Sea by applying the NLFT with quasi-periodic boundary conditions to the KdV equation. Our investigation revealed that, in all cases, some smaller solitons were additionally associated with a rogue wave. Typical values of the amplitude of the second largest soliton $A_S^2$ are 20 %–30 % of $A_S^1$. The amplitude of the third largest attributed soliton $A_S^3$ is typically 10 %–20 % of $A_S^1$.

So far, the results show that high soliton amplitudes in the spectrum are associated with high absolute values of wave heights or crests. However, this does not necessarily imply that high solitons play a role in forming individual waves that are exceptional with respect to the surrounding wave field. To be able to compare different measurement samples, the soliton amplitudes $A_S^1$ were normalised by the significant wave height $H_s$ of the corresponding sample. By relat-

ing the normalised soliton amplitudes to the different time series categories, the importance of solitons for the relative height of rogue or maximum waves was investigated (Fig. 9). If solitons are to play a major role in the presence of rogue waves, their normalised amplitudes are expected to increase from non-rogue wave samples with $H$ $(H_s)^{-1} < 2.0$ through height and double rogue waves $(2.0 \leq H$ $(H_s)^{-1} < 2.3)$ to extreme rogue waves $(H$ $(H_s)^{-1} \geq 2.3)$. In fact, the median values of $A_S^1$ $(H_s)^{-1}$ are higher for rogue wave samples than for non-rogue wave samples, meaning that the distributions calculated from the rogue wave samples are shifted towards higher normalised soliton amplitudes with respect to the distribution calculated from non-rogue wave samples (Fig. 9). Additionally, the rogue wave sample distributions, especially those calculated from crest and extreme rogue samples, show heavier tails. The differences in the distributions suggest that solitons play a role in rogue wave generation. It is striking that not only extreme rogue waves but also crest rogue waves had a tendency to be associated with higher solitons. This makes sense when recalling that a soliton is not an oscillating wave and, due to its shape, contributes more to wave

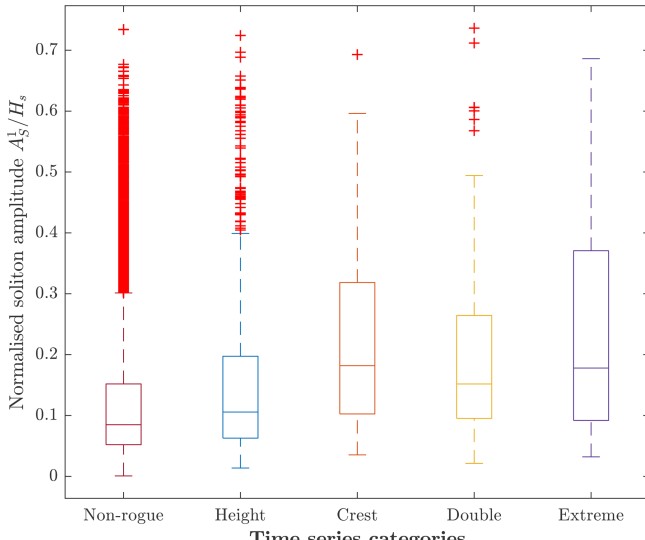

**Figure 9.** Amplitude of the highest soliton attributed to the rogue wave or maximum wave in the time series, normalised by the significant wave height, for the different categories of time series. Distributions are shown as box-and-whisker plots (box: interquartile range; whiskers: 1.5 times the interquartile range; horizontal line inside the box: median; red crosses: data outside the whiskers).

crests than to wave heights. However, although differences in normalised soliton amplitudes $A_S^1 (H_s)^{-1}$ are present for the different categories, the distributions overlap and the positive trend with increasing relative wave height is not as pronounced as the positive trend of $A_S^1$ with increasing maximum wave height (as presented in Fig. 8). This emphasises the relevance of the considered sea state for the soliton amplitude, in that large solitons are only found in high sea states. Large solitons correspond to high wave heights $H$ and high crest heights $C$ but not necessarily to high relative wave heights $H (H_s)^{-1}$ or high relative crest heights $C (H_s)^{-1}$.

As we were interested in the importance of nonlinear processes in rogue wave generation at the buoy location, we intended to quantify the nonlinearity of the rogue waves. In shallow water, the nonlinearity of waves can be described by the Ursell number (Ursell, 1953). According to Osborne (2010, Eqs. 10.151 and 10.154), the Ursell number in its time-like form is given by

$$Ur = \frac{3}{32\pi^2}\left(\frac{HL^2}{h^3}\right) = \frac{mK^2(m)}{2\pi^2}, \tag{13}$$

with the modulus $m$.

*Remark*: TS2 Different definitions of the Ursell number exist. A common definition is (Dean and Dalrymple, 1991, Eq. 11.109)

$$U_1 = \frac{HL^2}{h^3} = \frac{16}{3}K^2k^2, \tag{14}$$

with $K$ the complete elliptic integral of the first kind and with the modulus $k$. Comparison of Eqs. (13) and (14) shows the

Ursell numbers to differ by a factor of $3/(32\pi^2)$. The moduli in Eqs. (13) and (14) are related by $m = k^2$. Thus, different Ursell number definitions will yield different thresholds for the separation of wave theories. In this study, we use the definition given in Eq. (13) and adjust the cited threshold values accordingly. (For consistency with Eq. 14, the wave amplitude $a$ in the original equation of Osborne (2010) has been replaced by the wave height $a = H/2$.)

The Ursell number has been used to classify wave types. In Brühl (2014), solitary-like waves are defined by a modulus of $m > 0.99$. According to this classification and by applying Eq. (14), Ursell numbers $Ur > 0.559$ are obtained for solitary-like waves. Waves with $Ur \leq 0.559$ are classified as oscillatory waves.

According to Eq. (14), the Ursell number is defined either by the modulus $m$ or by height $H$ and wavelength $L$ of a single wave oscillation over depth $h$. Thus, we can calculate the Ursell number for the identified rogue waves using the $H$ and $L$ obtained by zero-upcrossing. In our case, the amplitudes of the largest attributed solitons show an almost linear positive trend with increasing Ursell number up until approximately $Ur = 0.5$ (Fig. 10). For our data, in which the bulk of waves are located below $Ur = 0.559$, this means that most rogue waves are not classified as solitons. This is in agreement with several previous studies that have shown that rogue waves in shallow water, despite their large amplitudes, have very small ratios of nonlinearity to dispersion (Ursell numbers) and, thus, are almost linear (Pelinovsky et al., 2000; Kharif and Pelinovsky, 2003; Pelinovsky and Sergeeva, 2006). Another observation made from Fig. 10 is an upper limit in soliton amplitude between $A_S^1 = 2.0$ m and $A_S^1 = 2.8$ m, depending on the time series category, for Ursell numbers larger than approximately $Ur = 0.5$. Referring to the classification given above, this implies that soliton amplitudes are limited for the most nonlinear waves, which are those satisfying solitary wave theory. A limit in soliton height as a result of breaking is expected at amplitudes of approximately $A = 8$ m for a water depth of $h = 10$ m, as the breaking criterion for solitary waves is $A h^{-1} = 0.78$ (McCowan, 1891) or $A h^{-1} = 0.83$ (Lenau, 1966). Therefore, shallow-water wave breaking at the location of the buoy can be excluded. The reason for the limit in soliton amplitude at $A_S^1 = 2.5$ m to $A_S^1 = 3$ m could be limited energy input by wind (see Middleton and Mellen, 1985, for soliton generation by wind) or a shoal in front of the measurement buoy causing the larger waves to break before they reach the buoy.

## 3.2 Soliton spectra for time series with and without rogue waves

When investigating the attribution of solitons to rogue waves in Sect. 3.1, we found that the largest soliton in the nonlinear spectrum could be attributed to the rogue wave in the majority of cases. In addition, this soliton was often outstanding from the other solitons in the spectrum, with a much

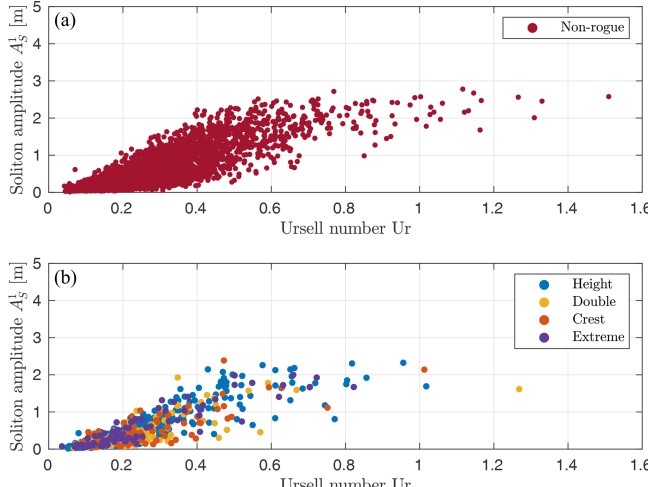

**Figure 10. (a)** Amplitude of the highest soliton attributed to the maximum wave in the time series as a function of the Ursell number of the maximum wave in the time series. **(b)** Amplitude of the highest soliton attributed to the rogue wave as a function of the Ursell number of this rogue wave.

larger amplitude than the remaining solitons in the spectrum (see the example in Fig. 6). Therefore, we were interested in whether the existence of an outstanding soliton in the nonlinear spectrum was typical of rogue wave samples off Norderney. We investigated this question statistically by comparing soliton spectra, calculated from vKdV-NLFT, for non-rogue wave samples and the four different categories of rogue wave samples. In fact, while all 15 156 considered time series yielded discrete spectra with a large number of solitons, we identified two characteristic classes of soliton spectra. The typical appearance of a soliton spectrum calculated from a time series without rogue waves was a cluster of solitons (Fig. 11). On the contrary, in the majority of cases, soliton spectra calculated from time series including a rogue wave showed one outstanding soliton with an amplitude much larger than that of the remaining cluster of solitons in the spectrum (Fig. 5).

To distinguish between clustered soliton spectra and those featuring an outstanding soliton, we compared the amplitudes of the largest soliton, $A_1$, and the second largest soliton, $A_2$, in the discrete spectrum. From the visual inspection of the spectra, we identified a threshold of the ratio $A_2 (A_1)^{-1}$, below which the largest soliton could be called outstanding:

$$\frac{A_2}{A_1} \leq 0.8. \tag{15}$$

Thus, a soliton spectrum had an outstanding soliton if the second largest soliton was at least 20 % smaller than the largest soliton in the spectrum. The choice of this threshold was further supported by the fact that the threshold $A_2 (A_1)^{-1} = 0.8$ coincides with the median value of $A_2 (A_1)^{-1}$ for maximum wave heights just below the rogue

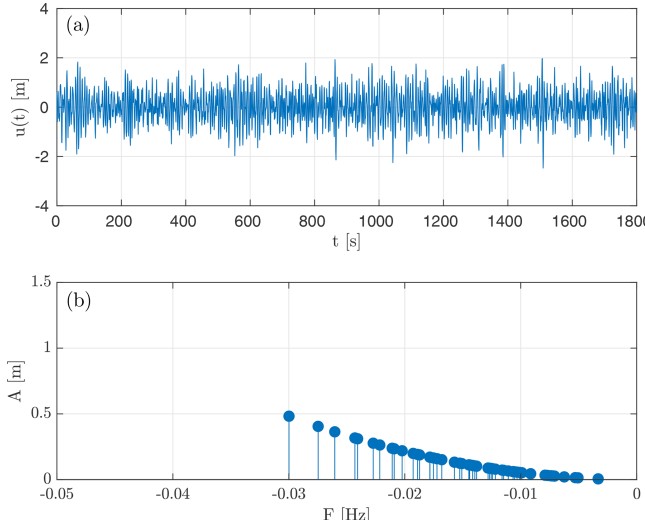

**Figure 11.** Example of **(a)** a non-rogue wave time series without rogue waves and **(b)** its corresponding soliton spectrum calculated from vKdV-NLFT. The soliton spectrum displays a cluster of solitons, found to be typical of the majority of spectra calculated from non-rogue wave time series. The time series was measured on 26 December 2016, starting at 11:30 CEST, with the parameters $H_{\max} = 4.44$ m, $H_s = 2.46$ m and $H_{\max} (H_s)^{-1} = 1.80$.

wave criterion $H (H_s)^{-1} \geq 2.0$ (Fig. 12). This reveals that our threshold chosen for the distinction between clustered spectra and those featuring an outstanding soliton concurrently indicates a difference between the spectra calculated from non-rogue and those calculated from rogue wave time series.

Equation (15) is valid for 30 min samples at the measurement site, which is the standard window size of measurement samples delivered by Datawell Waverider buoys. As the ratio between soliton amplitudes might be dependent on the window size, it is not clear if Eq. (15) would apply to time window sizes other than 30 min. The effect of a larger time window size will be discussed in Sect. 4. Table 2 shows the share of outstanding solitons and clustered soliton spectra in each of the categories defined in Sect. 2.1. It is seen that the typical appearance of the soliton spectrum for 30 min wave measurement samples off Norderney without rogue waves is a cluster of solitons (64 % of the samples); at the same time, it is not unlikely to obtain a soliton spectrum with one outstanding soliton from vKdV-NLFT (36 % of the samples). For 30 min rogue wave samples, in contrast, it is more likely to obtain a soliton spectrum with one outstanding soliton than a clustered soliton spectrum. This is true for height rogue samples (57 %), and it is even more pronounced for crest rogue samples (64 %), double rogue samples (72 %) and, finally, extreme rogue samples (87 %). The conclusion can be drawn that the absence of an outstanding soliton is a strong indicator of the absence of an extreme rogue wave. The differences between the four rogue wave categories, indicating that the

**Table 2.** Share of samples in each category showing an outstanding soliton or a clustered soliton spectrum, respectively.

| | Non-rogue | Height rogue | Crest rogue | Double rogue | Extreme rogue |
|---|---|---|---|---|---|
| Outstanding soliton | 36 % | 57 % | 64 % | 72 % | 87 % |
| Clustered solitons | 64 % | 43 % | 36 % | 28 % | 13 % |

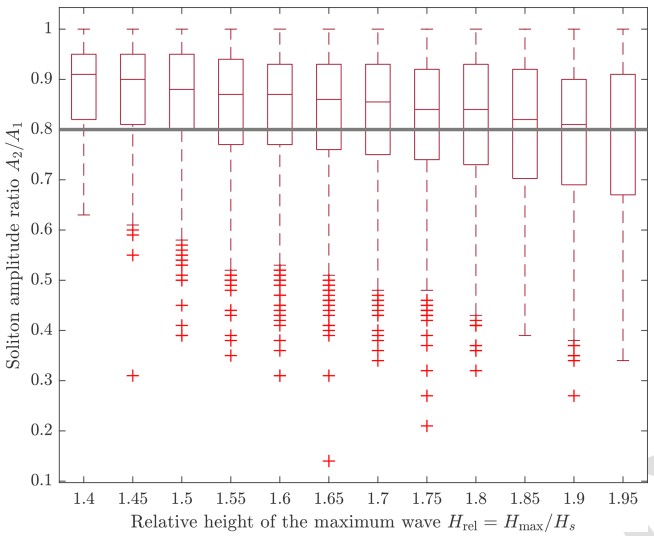

**Figure 12.** Distribution of the ratio between the second largest and the largest soliton in the discrete spectrum calculated from non-rogue wave time series. $H\,(H_s)^{-1}$ bins of width 0.05 are shown up until $H\,(H_s)^{-1} < 2.0$, which corresponds to the definition of height rogue samples (Eq. 4). Distributions are shown as box-and-whisker plots (box: interquartile range; whiskers: 1.5 times the interquartile range; horizontal line inside the box: median; red crosses: data outside the whiskers).

presence of an outstanding soliton is not equally expressive for all types of rogue waves, may lead to the presumption that not all rogue waves found off Norderney can necessarily be explained by the same theory.

The question regarding whether inferences can be made from the time to the spectral domain or vice versa is answered by a contingency table (Fig. 13). Here, all previously defined rogue wave categories are combined into one joint group of rogue wave samples. Two statements can be made based on the table. On the one hand, the probability that an NLFT spectrum calculated from a normal sample shows an outstanding soliton is $4986/13.984 = 36\%$, whereas the probability that a spectrum calculated from a rogue wave sample shows an outstanding soliton is $726/1172 = 62\%$. This indicates that, although not all rogue waves can necessarily be explained by the same theory, outstanding solitons occurred in connection with the majority of observed rogue waves off Norderney. While outstanding solitons play a role in 62 % of the cases in the combined group of rogue waves, the share

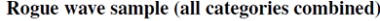

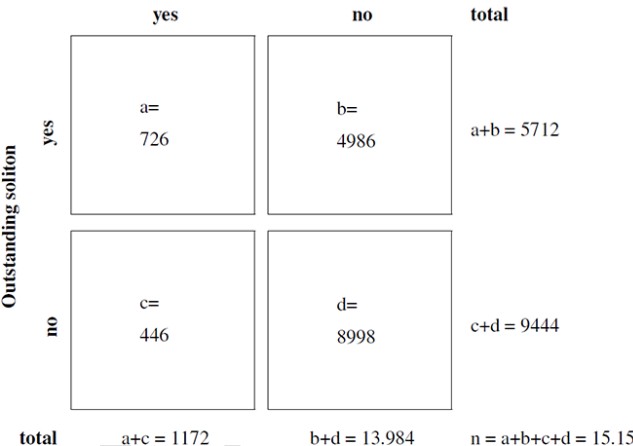

**Figure 13.** Contingency table of forecast–event pairs. The letters used in the table denote the following: $a$ – hits, $b$ – false alarms, $c$ – misses and $d$ – correct negatives.

differs between the rogue wave categories (Table 2). On the other hand, although rogue waves are more likely to be observed when an outstanding soliton is present in the NLFT spectrum, the presence of an outstanding soliton alone is not a sufficient an indicator for the detection of rogue waves. The main difficulty is the imbalance in sample size between non-rogue wave and rogue wave samples.

In Fig. 14, the ratio between the amplitudes of the second largest and the largest soliton in the nonlinear spectrum, $A_2\,(A_1)^{-1}$, is visualised in a box plot for each of the time series categories. A ratio above $A_2\,(A_1)^{-1} = 0.8$, meaning that the second largest soliton has a rather similar amplitude to the largest soliton, implies that the soliton spectrum is clustered (Eq. 15). For non-rogue wave samples, this is the case for the bulk of time series. The median of the ratio $A_2\,(A_1)^{-1}$ decreases from the leftmost to the rightmost category on the right axes in Fig. 14. For height rogue samples, the median of $A_2\,(A_1)^{-1}$ is below the 80 % line, with the distribution extending above and below. For double and extreme rogue waves, the gap between the soliton amplitudes may become much larger than for height rogue waves. In some cases, the amplitude $A_2$ amounts to less than 30 % of the amplitude $A_1$. In all categories except extreme rogue samples, there are samples for which the first and second solitons are almost similar in amplitude ($A_2\,(A_1)^{-1} \approx 1$). On the contrary, for

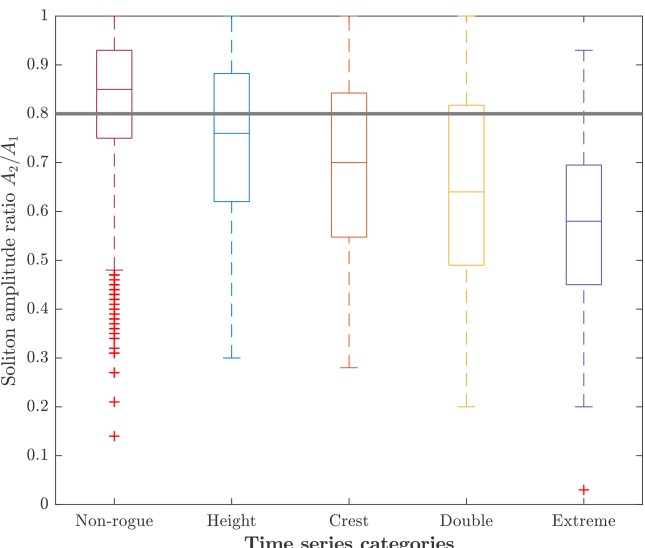

**Figure 14.** Box plots of the ratio between the second largest soliton ($A_2$) and the largest soliton ($A_1$) in the spectrum for the different categories of time series. Distributions are shown as box-and-whisker plots (box: interquartile range; whiskers: 1.5 times the interquartile range; horizontal line inside the box: median; red crosses: data outside the whiskers). Below the horizontal line denoting 80 %, the highest soliton in the spectrum is classified as outstanding.

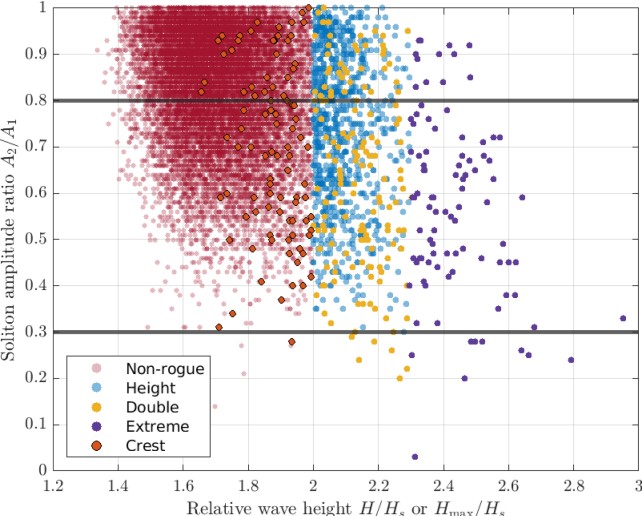

**Figure 15.** Ratio between the second largest soliton ($A_2$) and the largest soliton ($A_1$) in the spectrum as a function of relative wave height $H \, (H_s)^{-1}$ or $H_{\max} \, (H_s)^{-1}$ for the different categories of time series. Below the horizontal line denoting 80 %, the highest soliton in the spectrum is classified as outstanding. Below the horizontal line denoting 30 %, the highest soliton in the spectrum is referred to as strongly outstanding.

all extreme rogue wave samples, $A_2$ is below 93 % of $A_1$. The large part of soliton spectra from extreme rogue samples shows an outstanding soliton.

Figure 15 presents the ratio $A_2 \, (A_1)^{-1}$ in a scatter plot with one data point for each individual time series. According to this representation, although the presence of an outstanding soliton with $A_2 \, (A_1)^{-1} \leq 0.8$ is not a useful indicator of whether a rogue wave is present in the time series or not, the presence of a rogue wave becomes much more likely when one soliton in the nonlinear spectrum is strongly outstanding with $A_2 \, (A_1)^{-1} \leq 0.3$: of all 23 samples satisfying $A_2 \, (A_1)^{-1} \leq 0.3$, only $4/23 = 17$ % are non-rogue wave samples, whereas $19/23 = 83$ % of the samples are rogue wave samples (1 height, 1 crest, 8 double and 9 extreme rogue samples).

## 4    Discussion

We investigated discrete nonlinear soliton spectra obtained by the application of the vKdV-NLFT to time series measured by a surface-following buoy off the coast of the island of Norderney in the southern North Sea. The impulse to investigate the data at this specific site using nonlinear methods was given by a previous study (Teutsch et al., 2020). In the aforementioned publication, it was found that, while the Forristall distribution was sufficient to describe rogue wave occurrences at nearby buoy stations in somewhat deeper wa-

ter (see $kh$ ranges of buoy stations in Table 1 of Teutsch et al., 2020), the Norderney buoy recorded a larger number of rogue waves than expected according to the Forristall distribution. The results described in this paper suggest that nonlinear processes may explain the enhanced rogue wave occurrence at this specific site. The results were derived by the application of vKdV-NLFT and are, therefore, strictly valid for shallow-water conditions in the context of the applicability of the KdV equation. In a future study, it may be interesting to extend the investigation to additional sites with shallow water depths.

Throughout the study, indications were found that, although solitons play a role in the presence of rogue waves at Norderney, the soliton spectrum alone does not yield a satisfactory explanation of the formation of extreme waves/crests. A first hint is given in Fig. 5a, which shows the reconstructed soliton train along with the measured time series. Here, solitons (and their interactions) neither account for the full height of the observed rogue wave nor provide the observed wave trough. Figure 8 supports the finding that the solitons were not large enough to explain the full heights of the associated rogue waves. From Fig. 9, it is seen that the presence of a large soliton is not necessarily connected to the presence of a rogue wave. In addition, Kharif and Pelinovsky (2003) found that the interaction of unidirectional KdV solitons does not result in exceptional increases in wave elevation. As a consequence, one may speculate that the formation of the rogue waves in our dataset was a result of nonlinear interactions of one or more solitons with the underlying oscil-

lating wave field. This hypothesis will need further analyses to be validated.

The bathymetry below the measurement buoy at Norderney is characterised by a strong decrease in water depth. Non-Gaussian wave characteristics as a result of decreasing water depth have already been described by studies such as Huntley et al. (1977) in the context of wave run-up. It has gained increased attention in the context of rogue wave occurrence (e.g. Sergeeva et al., 2011). Increased rogue wave frequencies behind slopes or steps have been confirmed by numerous numerical (e.g. Sergeeva et al., 2011; Majda et al., 2019) and experimental studies (e.g. Trulsen et al., 2012; Kashima et al., 2014; Ma et al., 2014; Raustøl, 2014; Jorde, 2018; Bolles et al., 2019; Zou et al., 2019; Zhang et al., 2019; Trulsen et al., 2020). The main subject that the mentioned studies are concerned with is that waves propagating over a slope, step or bar are forced into new equilibrium conditions (Zeng and Trulsen, 2012). This mechanism is associated with strong non-Gaussian statistics and an increased rogue wave probability (Zhang and Benoit, 2021). The reason for the enhanced rogue wave probability was identified as the higher degree of nonlinearity in the shallow water behind the slope or step, which leads to an enhancement of second-order harmonic-bound waves (Gramstad et al., 2013). Zheng et al. (2020) and Li et al. (2021) confirmed (numerically and theoretically) that second-order terms (made up from bound waves and free waves released by the interaction of bound waves with the slope) are responsible for peaks in skewness and kurtosis. Zhang and Benoit (2021) stated that both second- and third-order effects evolving from the non-equilibrium dynamics at the depth transition significantly enhance the local kurtosis and the occurrence of rogue waves. For these effects to occur, the shallow domain must be sufficiently shallow, and the slope of the bathymetry change plays a major role (Fu et al., 2021). The largest peaks in kurtosis and skewness and the highest rogue wave probabilities were found for the steepest slopes (Gramstad et al., 2013; Zheng et al., 2020; Fu et al., 2021; Lawrence et al., 2021). Using tank experiments, Doeleman (2021) recently showed that the effect of slope is weakened in shallow water. Mendes et al. (2022) confirmed theoretically that a strong amplification may be found in intermediate water ($0.5 < kh < 1.5$). They stated that "Whether rogue waves are enhanced in strong bathymetry changes throughout most oceans or regionally under suitable conditions is yet to be assessed" (Mendes et al., 2022). Zeng and Trulsen (2012) anticipate that the described mechanisms may explain the spatially varying occurrence frequency of rogue waves on the continental shelf, where waves enter from the deep sea. Therefore, the described processes associated with a strong decrease in depth might be an explanation for the observed increased rogue wave occurrence off the coast of Norderney (Teutsch et al., 2020). A connection between rogue waves and solitons in this context was established by Sergeeva et al. (2011). The authors showed, by applying a KdV equation,

**Table 3.** Share of samples in each category showing an outstanding soliton in the soliton spectrum, for the respective water depth adopted in the NLFT calculation. Note that the shallow-depth criterion in Eq. (3) changes to $T_p > 5$ s for a water depth of $h = 12$ m, which left approximately 94 % of the samples for the calculation at a water depth of 12 m.

| Water depth | Non-rogue | Height rogue | Crest rogue | Double rogue | Extreme rogue |
|---|---|---|---|---|---|
| 8 m | 32 % | 57 % | 61 % | 73 % | 75 % |
| 10 m | 36 % | 57 % | 64 % | 72 % | 87 % |
| 12 m | 36 % | 53 % | 62 % | 70 % | 76 % |

that the number of solitons increases in the shallow water behind a slope. They linked this increased soliton occurrence to an increased rogue wave probability.

The solutions of the KdV equation for a given free-surface elevation time series strongly depend on the water depth (see Eq. 7). While we assumed a constant water depth of $h = 10$ m for our calculations, there are in fact major uncertainties regarding the water depth at the actual location of the buoy, due to tidal changes and bathymetry gradients as well as the movement of the buoy, as mentioned in Sect. 2.1 (Fig. 2). The mean tidal range at Norderney is approximately 2.5 m; however, due to an additional movement of the buoy of 2 m to each side of the slope, a total deviation from the nominal water depth of $\pm 2$ m is reasonable. We performed a sensitivity analysis to test the robustness of the results with respect to these uncertainties. To do so, we repeated the computation of the soliton spectrum for water depths of $h = 8$ and 12 m, respectively, while using the same free surface data as in the previous analysis. A changed water depth leads to a different depth range in which the KdV equation is valid (Eq. 3). For the calculation with a depth of $h = 12$ m, we repeated the identification of the samples that fulfil shallow-water conditions in the KdV context, as samples and maximum waves (due to the larger water depth) now had to satisfy the condition $T_p$ or $T > 5$ s in order to classify as shallow-depth samples/waves for the applicability of the KdV equation. Therefore, only 14 206 samples (i.e. approximately 94 % of the original sample size) were available for the calculation at $h = 12$ m. For the calculation with a depth of $h = 8$ m, we used the same samples as for the calculation with $h = 10$ m, as these automatically fulfilled shallow depth conditions at $h = 8$ m. Irrespective of the water depth adopted in the calculation, the result remained that samples with rogue waves, especially extreme rogue waves, were more likely to contain an outstanding soliton in the nonlinear spectrum than samples without rogue waves (Table 3). Thus, the results are robust with respect to potential uncertainties in water depth.

The KdV equation is only valid for unidirectional waves. Although Osborne (1993) recommends the application of the NLFT for KdV to measurement data only for samples in which the largest part of the energy is in the dominant propa-

**Table 4.** Share of samples in each category showing an outstanding soliton, for the approximately 10 % of samples with the lowest directional spreading.

|                      | Non-rogue | Height rogue | Crest rogue | Double rogue | Extreme rogue |
|----------------------|-----------|--------------|-------------|--------------|---------------|
| No. of samples       | 1614      | 91           | 12          | 17           | 10            |
| Outstanding soliton  | 31 %      | 57 %         | 67 %        | 88 %         | 90 %          |

gation direction, we applied the KdV-NLFT outside the limits that are given in the literature. At our measurement site, the sea state was always multidirectional, with a directional spreading of the wave energy approximately between 28 and 55°, whereas only 5 % of the energy was perpendicular to the dominant direction of propagation in the dataset of Osborne (1993). We repeated the first part of the analysis, for which the results are described in Sect. 3.1, for the approximately 10 % of samples in each category with the lowest directional spreading. This corresponded to a threshold in directional spreading of 35° for most categories, except crest rogue waves, which tended to occur in broader sea states (threshold at 36.5°), and extreme rogue waves, which statistically occur in more narrow sea states (Christou and Ewans, 2014) (threshold at 34°). We found our result – that an outstanding soliton is more typical of a rogue wave time series than for a non-rogue wave time series – confirmed and partly emphasised (Table 4). Therefore, we rate vKdV-NLFT, although assuming unidirectionality in multidirectional measurement samples, an appropriate tool to evaluate the connection between solitons and rogue waves off Norderney.

In our study, we applied the vKdV-NLFT as a trace method for (extreme) rogue waves and demonstrated, for the first time, that certain distinctive patterns in the NLFT spectrum of real-world time series indicate extreme rogue waves. The method may provide further information on possibly dangerous time series in future applications. Further research is required on the applicability of the KdV equation to our data, which cannot be validated on the basis of single-point measurements. If wave propagation at Norderney is well described by KdV theory, the NLFT spectrum is approximately constant during propagation. The method may then identify time series with the potential of forming extreme rogue waves. Moreover, even if the KdV equation does not describe the propagation well, we still consider the NLFT a more appropriate transform than the linear FFT, which is often applied even if waves are nonlinear. Similar to the FFT in the linear case, our method should be treated as a signal transform (Sugavanam et al., 2019). Our study provides insights into the spectral characteristics at the considered site.

We would like to put an emphasis on the limitation of our suggested definition of an outstanding soliton (Eq. 15) to the size of the measurement window. Our criterion was chosen based on the inspection of soliton spectra from 30 min time series. However, the gap size might change depending on the chosen window size. An increase in window size, meaning more waves in the time series, will introduce additional solitons to the spectrum. If these are larger than $A_1$ or emerge in between $A_1$ and $A_2$, the gap size between the two largest solitons will be influenced. If these are smaller than $A_2$, their emergence will not alter the gap between $A_1$ and $A_2$. Similarly, a reduction in window size would exclude waves in the time series and remove solitons corresponding to these waves. If this modification leads to the removal of the largest or second largest soliton, the gap between the new $A_1$ and $A_2$ will become larger or smaller than for a 30 min time window. If this modification only affects solitons smaller than $A_2$, the size of the gap between $A_1$ and $A_2$ will not be influenced. We applied the ratio between $A_2$ and $A_1$ merely as a measure to statistically evaluate differences in the soliton spectra calculated from 30 min non-rogue wave and rogue wave time series. For different window sizes, it might be necessary to define new criteria.

Due to the limited recording frequency of the wave buoy, one might question the correct assignment of time series to the different categories (Table 1). Wave crests might be missed by the discrete measurement points, leading to a possible underestimation of rogue or extreme rogue samples (Stansell et al., 2002). However, even if extreme rogue time series were assigned incorrectly to the category of height rogue samples, this misinterpretation is conservative: none of the time series in the extreme rogue category has been assigned incorrectly. Furthermore, according to the sampling theorem (Shannon, 1949), the buoy sampling rate of 1.28 Hz is sufficient to sample time series whose FFT spectra decay at approximately 0.6 Hz (Fig. 5b). Therefore, we consider the buoy sampling frequency sufficient for our purpose.

Our result that rogue wave samples have a higher probability of showing an outstanding soliton in the nonlinear spectrum compared with non-rogue wave samples becomes most obvious in the categories of double and extreme rogue samples. In these categories, differences from non-rogue wave samples are visible not only in the percentage of outstanding solitons but also in the magnitude of the amplitude gap between the first and second solitons in the spectrum. Height rogue waves, on the contrary, do not seem to differ very much from high waves in non-rogue wave samples, both in terms of the gap between first and second soliton in the spectrum and in the height of the solitons associated with the maximum wave. The fact that differences between time series

with and without rogue waves become apparent only in some of the chosen categories raises questions regarding whether the choice of rogue wave definitions is reasonable for the considered location. The rogue wave definitions serving as a basis to this study were introduced by Haver and Andersen (2000) for deep-water waves. The relative height and crest values in their definitions represent outliers, being exceeded in 1 of 100 cases when applying a second-order model to the deep-water sea surface elevation (Haver, 2000). The definitions have been taken up numerous times in the literature. Authors have been investigating whether rogue waves according to the definition of Haver and Andersen (2000) are outliers with respect to typical wave distributions in the real ocean as well (e.g. Forristall, 2005; Gemmrich and Garrett, 2008). The question of whether rogue wave definition by a certain height or crest threshold is useful in practice has been raised (Häfner et al., 2021). Several authors have, based on large measurement datasets, come to the conclusion that these rogue waves are rare but are, nevertheless, realisations of commonly used wave distributions (e.g. Waseda et al., 2011; Christou and Ewans, 2014). In a previous study (Teutsch et al., 2020), we were able to confirm this conclusion at buoy measurement stations in intermediate water. However, at the buoy station off Norderney, in a comparably shallow water depth, that showed a larger number of rogue waves than expected according to the common wave distributions, the interaction of solitons with oscillating waves might be a mechanism explaining the increased occurrence of rogue waves.

## 5   Conclusions

Rogue wave occurrence recorded off the coast of the island of Norderney is not sufficiently explained by the Forristall distribution of wave heights. We investigated the role of solitons as components of the discrete vKdV-NLFT spectrum in the enhanced rogue wave occurrence. Our main results for this specific measurement site are as follows.

- Each measured rogue wave could be associated with at least one soliton in the NLFT spectrum.

- The soliton heights were always smaller than those of the rogue waves. Samples with rogue waves were more likely to contain an outstanding soliton in the NLFT spectrum than samples without rogue waves.

- The soliton spectrum analysis is a good indicator of extreme rogue waves in the corresponding time series.

- The presence of a strongly outstanding soliton, with a ratio between the second largest and the largest soliton in the nonlinear spectrum of $A_2 (A_1)^{-1} \leq 0.3$, was found to be a strong indicator for the presence of a rogue wave.

- Conversely, the absence of an outstanding soliton in the spectrum is a strong indicator for the absence of an extreme rogue wave of $H (H_\mathrm{s})^{-1} \geq 2.3$.

We conclude that nonlinear processes are important in the generation of rogue waves at this specific site and may explain the enhanced occurrence of such waves beyond common wave height distributions. Rogue waves at Norderney are likely to be a result of the interaction of solitons with the underlying field of oscillatory waves. The nature of this interaction should be subject to further research.

*Code availability.* The FNFT software library is available via https://doi.org/10.21105/joss.00597 TS3. The specific commit used for this work is available at https://github.com/FastNFT/FNFT/archive/681191c86eefbe4a570a9cf3a457577e7b75cc5e.zip TS4.

*Data availability.* The CE4 underlying wave buoy data are the property of the Lower Saxony Water Management, Coastal Defence and Nature Conservation Agency (NLWKN). They can be obtained upon request from the agency TS5.

*Author contributions.* All authors contributed to the idea and scope of the paper. IT performed the analyses and wrote the manuscript. MB, RW and SW provided help with data analysis, discussed the results and contributed to writing the paper. RW supervised the work.

*Competing interests.* The contact author has declared that none of the authors has any competing interests.

*Acknowledgements.* The buoy data were kindly provided by the Lower Saxony Water Management, Coastal Defence and Nature Conservation Agency (NLWKN).

*Financial support.* This CE5 project has received funding from the European Research Council (ERC) under the European Union's Horizon 2020 Research and Innovation programme (grant agreement no. 716669). Ina Teutsch received funding for this work from the Federal Maritime and Hydrographic Agency (BSH).

The article processing charges for this open-access publication were covered by the Helmholtz-Zentrum Hereon.

*Review statement.* This paper was edited by Ira Didenkulova and reviewed by two anonymous referees.

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

**Remarks from the language copy-editor**

CE1    Apologies for the oversight. Please confirm the change.

CE2    Please note that our house standards require the use of numerals in a mathematical context such as this.

CE3    Please confirm the change.

CE4    Please note the slight edit. The omitted information affected the readability of the sentence and is given in the Acknowledgements.

CE5    Please note the slight edit.

**Remarks from the typesetter**

TS1    Please note that these changes require approval from the handling editor. Please provide an explanation of why this needs to be changed that we can forward to the editor. The status of your paper will be changed to "Post-review adjustments" until the editor has made their decision. We will keep you informed via email. Thank you.

TS2    Thank you for the explanation. If the equation does not need to be numbered, the text can also be moved to a footnote. In this case, Eq. (14) would be unnumbered as part of the sentence and Eq. (15) would become Eq. (14). Please let me know if you would prefer this solution.

TS3    Please provide a direct link to the software code and, if possible, a DOI instead of a URL. In any case, please provide a reference list entry including creators, title, and date of last access.

TS4    Please clarify whether the code is your own. If yes, please provide a DOI in addition to your GitHub URL since our reference standard includes DOIs rather than URLs. If you have not yet created a DOI for your code, please issue a Zenodo DOI (https://help.github.com/en/github/creating-cloning-and-archiving-repositories/referencing-and-citing-content). If the code is not your own, please inform us accordingly. In any case, please ensure that you include a reference list entry corresponding to the code including creators, title, and date of last access.

TS5    Please indicate how the agency can be contacted.

TS6    Please provide date of last access.

TS7    Please provide a persistent identifier (ISBN or DOI preferred).

TS8    Please provide publisher and a persistent identifier (ISBN or DOI preferred).

TS9    Please provide a persistent identifier (ISBN or DOI preferred).

TS10    Please provide date of last access.

TS11    Please provide publisher.

TS12    Please provide a persistent identifier (ISBN or DOI preferred).

TS13    Please note that "[code]" could not be added as this is a journal reference.