# Peer review of "Contribution of solitons to enhanced rogue wave occurrence in shallow depths: a case study in the southern North Sea"

_Natural Hazards and Earth System Sciences, 2022_

## Referee Comment (RC1)

**Review**

On the manuscript **"Contribution of solitons to enhanced rogue wave occurrence in shallow water: a case study in the southern North Sea"** by Ina Teutsch, Markus Brühl, Ralf Weisse and Sander Wahls.

**Overview: Conditional Acceptance Upon Major Revision**
* * *
The paper presents a possible explanation of rogue wave statistics collected over six years in the coast of Germany, consisted of intermediate and shallow water regimes. The authors argue that given the failure of second-order models to correctly describe rogue wave statistics in their previous article (Teutsch et al., 2020), the interaction between solitons and the linear oscillatory wave components of entire time series might be a viable alternative theory/process, whose methodology relies on the analysis of the soliton spectrum. Clearly, the topic is of the highest relevance and of cutting-edge nature in ocean sciences, while also being within the scope of the journal. Provided major amendments are implemented, I believe the revised version of this paper would be an essential reading for everyone studying extreme waves in the ocean.

I would like to provide specific and general comments on three different types:

1. General comments that address references, explanation of scientific terms and further clarifications that are essential for the general reader to follow the scientific reasoning behind this work.

2. Suggestions that would improve the scientific quality of the manuscript.

3. Strong scientific issues in the manuscript that must be revised before I can recommend acceptance.

**1. General Comments**
* * *
I believe the reading quality of the paper is fair but can be significantly improved. Unfortunately, the organization, literature coverage and introduction of scientific terms could have been better implemented. Hence, the suggestions and requests follow:

1A. There is no introduction to what are rogue waves. **Please, introduce rogue waves and their relevance as a natural hazard properly.** For instance, design waves are necessary for the construction of the state-of-the-art vessels and offshore structures, and rogue wave likelihoods are essential for classification societies of ocean engineering (Bitner-Gregersen and Gramstad, 2015).

1B. In the beginning of the first paragraph it is stated:

*"There has been a lively discussion on whether the occurrence frequency of rogue waves in the open ocean is well described by second-order models. Both Rayleigh (Longuet-Higgins, 1952) and Weibull distributions (Forristall, 1978) have been used to describe the distributions of wave and crest heights."*

The referenced distributions are not of second-order[1]. It may be the case that this is simply a jump in the story being told in the introduction, and readers may believe these distributions are indeed of second-order. **Please add the references for the discussion about second-order models range of validity.** Given that these distributions are actually of first-order[2], **I suggest you clearly state so and preferably write this before you discuss second-order models.**
* * *
[1] Whenever second-order models are mentioned they mean second-order in steepness (Tayfun and Fedele, 2007).

[2] That is, for the superposition of linear waves (see section 2 of Longuet-Higgins (1952)).

1C. A few lines down the first paragraph, I find the statement:

*"Independent of the measurement device, some authors found measured wave heights to agree well with the established distributions, while others found the frequency of rogue wave occurrences over- or underestimated."*

**Please be specific and add the references for these distributions and studies.** While this is a correct statement, not every reader knows which studies you refer to. However, after this statement the authors provide a few examples. Unfortunately, these are very few examples among several dozen works. **It would be better to cluster all studies[3] known to the authors as agreeing (references), overpredicting (references), underpredicting (references).** Such a suggestion would avoid the following issues found in nearby statements as:

*"While Olagnon and van Iseghem (2000) found rogue wave occurrences to be overpredicted by the classical distributions."*

What are classical distributions? This is quite confusing for both experts and first readers of rogue wave research.

*"Rogue wave occurrences in buoy data from the US coast, recorded in shallow, intermediate and deep water, were found to be strongly overestimated by a Rayleigh distribution."*

Again, where is the reference? I assume this is Cattrell et al. (2018). Or is it the Baschek and Imai (2011) study cited one sentence later? **The discussion between lines 25 and 30 must be entirely rewritten. May I suggest that you explain when first-order distributions have different outcomes (agreement, overprediction, underprediction), then move on to second-order models, and always include the references for the models as well as for the studies assessing their predictions.**

1D. Lines 30-31 have a confusing statement from the grammar point of view:

*"Furthermore, the respective authors describe local differences in rogue wave occurrence frequency between their measurement stations (Baschek and Imai, 2011), depending on the wave climate and especially in coastal waters, where waves interact with the seabed (Cattrell et al., 2018; Orzech and Wang, 2020)."*

The use of "the respective" is in contradiction with the citep format. **I suggest you simply remove these two words.**

1E. You should probably add three further comments on large data sets and their conclusions:
Firstly, **that Karmpadakis et al. (2020) showed that distributions can perform well in a narrow range of sea parameters (steepness, bandwidth and so on) but no available model performs well in a wide range of sea parameters. In my opinion, this is the single most important finding in recent years.**
Secondly, **you should cite Häfner et al. (2021)[4] as another crucial real ocean study that provided unfavorable evidence for established models, including modulational instability.**
Thirdly, to address Karmpadakis et al. (2020), **Mendes and Scotti (2021) showed that a theoretical superposition of established complementary models that individually depend on steepness and water depth can describe the unexplained uneven distribution of rogue waves in deep and intermediate waters reported by Stansell (2004), that is, can describe why some sea states (steepness, depth, etc) have higher rogue wave frequency than others.**
* * *
[3]There is no need for an exhaustive list.
[4]They have cited this article, although at the discussion section and never mentioned its main results as described here.

1F. You should probably add some references after line 42 stating that **there is no continuous probability model that describes both deep water and shallow water at a wide range of sea states**[5] (**Massel**, **2017**; **Karmpadakis et al.**, **2020**).

1G. You should probably add some references after citing Benjamin and Feir (1967) on line 51 stating that **the BFI/Modulational instability do not perform well in real ocean time series (Fedele et al.**, **2016**; **Häfner et al.**, **2021**).

1H. You should be clear with the term "nonlinearity" on line 65 onwards. In many articles, nonlinearity is a code-name for either significant steepness or the Ursell number. **I suggest that when you speak of nonlinearity in general, you rewrite it as nonlinear processes**.

1I. You should add somewhere near the citation of Slunyaev et al. (2016) **that half a century earlier through depth-dependent breaking criteria Glukhovskii (1966) already predicted that rogue waves becomes less frequent than expected by Longuet-Higgins (1952)**, see Wu et al. (2016) for a review.

1J. Lines 78-79 have inconsistent comments:

*"However, so far only few studies have addressed the impact of bathymetry on rogue wave generation"*

This statement is in clear contradiction with the brief description of bathymetry effect on rogue waves at the discussion section of this very preprint, citing not less than a dozen articles since 2012. **I suggest rephrasing this comment in accordance with the original intent but having in mind the discussion section and the recommended further literature below.**

1K. Between lines 376-384 you should add several crucial references/comments:

**Numerical studies like Gramstad et al. (2013); Fu et al. (2021) showed that the slope of the bathymetry change plays a major role in rogue wave amplification. Moreover, Zheng et al. (2020); Lawrence et al. (2021) showed that increasing mild slopes increases the probability of amplification, whereas when slopes are too steep this effect is no longer noticed. Doleman (2021) recently further showed that the slope effect in shallow water is negligible, as opposed to intermediate water. Trulsen et al. (2020); Zhang and Benoit (2021) discussed that rogue wave amplification due to bathymetry only happens in a narrow range in water depth $0.5 < kh < 1.3$, whose theoretical explanation is provided by Mendes et al. (2022). Further evidence that when the water depth becomes too shallow throughout the shoaling process the amplification dies out is provided by Xu et al. (2021). You should also add other important experiments, such as: Raustøl (2014); Jorde (2018); Bolles et al. (2019); Zou et al. (2019).**

1L. The Bruhl (2022) reference simply could not be found. It appears to lead to a course page at TU Braunschweig.

**Scientific Improvement**
* * *
2A. Lines 32-38 seemingly contradict the main results of Teutsch et al. (2020):

*"In a previous study, we have analyzed measurement data from various stations in the southern North*
* * *
[5]See in particular the second paragraph of section 4.4.5 of Massel (2017).

*Sea (Teutsch et al., 2020) and found rogue wave frequencies[6] to vary spatially and by measurement device. For data obtained from wave buoy measurements, we generally found rogue wave frequencies slightly overestimated by the Forristall distribution,(...)"*

I am intrigued by this comment, because figures 7 and 9 (which show the exceedance probability for all data sets combined) clearly do not support that Forristall distribution is overestimating the observed statistics. In fact, the first paragraph of section 3.3 of Teutsch et al. (2020) states the opposite:

*"For wave heights up to twice the significant wave height, which corresponds to the threshold used to identify rogue waves, the measurement data are well described by the Forristall distribution. At a height of $H = 2H_s$, the data begin to deviate from the Forristall distribution. Both distributions increasingly diverge for larger relative wave heights, $HH_s^{-1}$ . This suggests that in our data, rogue waves occurred more frequently than could be expected from the Forristall distribution. The frequency of rogue waves much larger than twice the significant wave height also exceeded expectations given by the Rayleigh distribution."*

I can only believe this is a typo and the authors meant **that Forristall agreed with the statistics of large ordinary waves ($H < 2H_s$) but underestimated statistics of rogue waves ($H \geqslant 2H_s$). Please fix this contradiction.** Moreover:

*"An exception was one measurement buoy, which was located in the shallow waters off the coast of the island Norderney, Germany (Fig. 1). For this buoy, enhanced rogue wave occurrence, which could not be explained by the Forristall distribution, was observed."*

It is in contradiction with the previous text from the author's last paper based on the same data set.**The whole text has to be changed, as not just Norderney buoy showed much higher statistics than expected by Forristall (1978)[7]. This previous statement from the paper is problematic because it suggests that only shallow water statistics were not explained by Forristall and needs a new model, while Teutsch et al. (2020) showed that most stations (buoy or radar) feature the same underprediction by Forristall. I recommend the authors to analyze all stations where Forristall and Rayleigh underestimated rogue wave statistics, not only Norderney.** However, the Norderney sample contains enough waves to provide a statistically significant study of rogue waves in shallow water[8], so it is acceptable for the authors to focus their efforts on this data set alone, **provided they explain why this data set is representative of all others where the Forristall model underestimated rogue waves.**

2B. Page 20 introduces the Ursell number written in a very strange way, where it denotes the wave celerity by $c$ while a capital counterpart $C$ as wave crest. **I suggest you add a footnote explaining that many authors define Ursell as $Ur = HL^2/h^3$ and has a different scale for wave theories: for instance Dean and Dalrymple (1984) says Stokes wave must obey $Ur \leqslant 8\pi^2/3$. Your definition would be better written as $3/32\pi^2(HL^2/h^3)$ in which Stokes waves would find $Ur \approx 1/4$, in agreement with your discussion.**

2C. Again as in 2A, the comments on lines 368-370 find no agreement with the results published in Teutsch et al. (2020):

*"The impulse for investigating the data at this specific site by using nonlinear methods was given by a previous study (Teutsch et al., 2020). There, it was found that while second-order distributions were sufficient to describe rogue wave occurrences at nearby stations in somewhat deeper water, the Norderney buoy*
* * *
[6]Frequency is understood as the likelihood of appearance, as described in the caption of figure 7 onwards of Teutsch et al. (2020).

[7]Note that when distributions were separated by location in figure 8 of Teutsch et al. (2020), the buoy data of station WHS is in agreement with Forristall, while station LTH is underestimated by the latter.

[8]Rogue wave return periods according to Longuet-Higgins (1952) are of 1 RW per 2,980 waves, hence a data set (Norderney) with 15,000 samples of 30 minutes with mean period of $T_z \sim 5$s means a total of 5.4 million waves.

*recorded a larger number of rogue waves than expected according to second-order theory"*

Where in Teutsch et al. (2020) good agreement with Forristall (1978) in deeper waters is shown? It is also not of second-order. **On the contrary, table 1 of Teutsch et al. (2020) clearly shows that the stations named AWG and Clipper ranged from intermediate to deep waters, while WES and LTH remained mostly in intermediate waters[9]. Then in figure 9 Teutsch et al. (2020) delineates exceedance probabilities, of which AWG and Clipper in deeper waters are by no means in agreement with Forristall, and are actually more underpredicted by this distribution than the stations in shallow water.**

2D. The following statement on lines 374-376 is very misleading:

*"Non-Gaussian wave characteristics as a result of decreasing water depth have already been described e.g. by Huntley et al. (1977) and gained increased attention in the context of rogue wave occurrence."*

I have no idea why this reference appeared here. This text discusses no wave statistics whatsoever (I downloaded and read it), the only non-Gaussian characteristics is the run-up (important for some hazards but not for rogue waves) and the distribution is of run-up velocities, not of wave height measurements. They show the spectrum of run-up instead of a spectrum of wave amplitudes. In fact, the references of this article mentions papers related to swash, set-down, set-up and run-up over sloping beaches. Yet it does not mention to any previous wave statistics article such as Rice (1945); Longuet-Higgins (1952); Cartwright and Longuet-Higgins (1956); Longuet-Higgins (1963); Draper (1964); Glukhovskii (1966); Draper (1971); Jahns and Wheeler (1973); Mallory (1974); Earle (1975); Longuet-Higgins (1975); Haring et al. (1976) among others, and not cited by any relevant rogue wave review paper, which proves it has no connection with a rogue wave context. **I recommend the authors to remove this reference.**

**Scientific Issues**

The manuscript seems to contradict itself several times, often arguing that data suggests that large outstanding solitons are associated with rogue waves, only as a few lines below (found in several pages) discredit this suggestion. The conclusions do not reflect the discussion within the bulk of the text where it is argued that the soliton spectrum clearly can not account for the rogue wave statistics by itself. The reader would understand from the introduction and conclusions that the soliton spectrum is enough to predict rogue waves. The authors do not discuss or compute exceedance probabilities in the context of solitons, which diminishes the impact of the manuscript from an engineering and prediction perspective. Below I give more concrete constructive criticism:

3A. Despite discussing waves in shallow water and in a bathymetry with steep slope (line 138 and figure 2), the authors simply did not show values of water depth $kh$ nor the slopes $\nabla h$. **Please, provide this essential information. I should remind the authors that the definition of shallow water reads $h < L/20$ (Dingemans, 1997), or alternatively $kh < \pi/10$. I am not sure that all these data sets are actually within its mathematical definition of shallow water[10]. Moreover, KdV is restricted to slightly higher dimensionless depths of $kh \lesssim 1.4$ or $h/L < 0.22$ as the authors wrote it[11], but it is not the KdV that defines what the shallow water regime is. The authors should add a reference. Furthermore, the title should be rewritten as to convey either "Shallow Depths" or "Coastal Areas" that are consistent with the $kh$ range.**
* * *
[9]As defined by the shallow water limit $kh < \pi/10$.

[10]Actually, from table 1 of Teutsch et al. (2020) they are technically not in shallow water, and some stations have a lower range close to but not yet in shallow water.

[11]This can be found on page 559 of Dingemans (1997).

3B. On line 156 the authors refer to the wave celerity in shallow water as $c = \sqrt{gh}$, but as stated above, this formula is valid only for $kh < \pi/10$, whilst the authors use $kh < 1.38$ as the definition. **These two definitions are incompatible, which would affect the threshold for the peak period on line 159**. Although I understand that the 98% percentage of cases would drop to probably 90%, from the physical point of view you must be careful with definitions.

3C. Sample definitions ranging from "normal" to "extreme" on lines 166-186 are very problematic. **First and foremost, for engineering purposes a wave that exceeds a significant wave height of 15m (typical storm in the North Sea near Norway) by a 1.9 factor (28.5 m) is as dangerous as a rogue wave that exceeds it by a factor of 2 (30.0 m)! Secondly, as you may see on figure 14 of** Zhang and Benoit (2021)**, nonlinear processes such as the shoaling of second-order waves will increase the likelihood of all large waves that are not strictly defined as rogue waves** ($1.25 < H/H_s < 2$)**, so that it is highly biased to consider wave samples with maximum height** $H/H_s = 1.8$ **to be normal. Nonlinear processes are not restricted to waves with** $H/H_s > 2$**. Please change your text and definition accordingly.**

3D. Statistics based on samples can be misleading. **I suggest the statistical analysis to also include the total wave count/exceedance distribution as a complement. In this particular case, I suggest adding a new line in table 1 for the total count of waves and rogue waves according to each definition on page 11.**

3E. The article clearly draws the theories/methods developed by soliton interaction with linear waves in all the literature cited. Since the manuscript itself never conclusively claims what surface elevation they are actually covering, I would highlight the most important sources as I read in the manuscript: Osborne et al. (1991) and Bruhl and Oumeraci (2016). The results of Osborne et al. (1991) seem to be the core of the meaning of this paper: Solitons are hidden in the time series signal and can be detected with the NLFT. Additionally, Bruhl and Oumeraci (2016) argues that in depths not exceeding $h/L < 0.22$ cosine waves (or any oscillatory wave) are actually a transient wave composed by multiple solitons that mutually interact and at some point may disintegrate into separate solitons. Then, I can not understand why very little of this theory in these two articles are discussed in the introduction, without further expansion in the methods section. It took me quite a while to understand the relevance of the remainder of the manuscript without reading these two sources. Hence, the manuscript is not self-contained. **I believe the authors have to state the theory in these two articles very clearly and not briefly. Foremost, please make similar figures as 1a and 1d of** Osborne et al. (1991)**, to show the non-expert what you actually mean by soliton spectrum (side-by-side with the linear spectrum like 1d) and how solitons are "associated" with rogue waves (like figure 1a). Second of all, please state mathematically how you handle/describe the sea surface elevation components as in equation 6 of** Bruhl and Oumeraci (2016)[12] **with a similarly robust explanation of its meaning.**

3F. Let us assume that the soliton spectrum can single out rogue waves, and describe its formation through soliton interaction. The way I see it, the authors have overlooked the prediction/warning aspect of rogue wave research, which is intrinsic to all natural hazards. The eventual inability of the soliton spectrum method to become predictive does not diminish the importance of the manuscript, but it has to be stated. **I therefore suggest that the authors should outline how this process could become actually predictive for engineering purposes, in line with the journal scope. It is not at all clear to me how extracting the soliton spectrum of a time series (as times go) can predict rogue waves, that is, this seems to be a trace method. Can we infer the soliton amplitudes before the actual time series is recorded, for instance, from sea state variables? Regardless of the author's answers to these questions, this discussion must be included at the end of the manuscript.**

3G. Lines 217-223 on page 13 and all discussion following the soliton spectrum are all problematic:
* * *
[12]I wonder if we also could write the interaction as $\zeta(x,t) = \zeta_{linear} + \zeta_{second-order} + \zeta_{soliton}$, with $\zeta$ being the sea surface elevation, because at $0.05 < h/L < 0.22$ second-order waves may still exist.

*"Solitons were found in all samples, with and without rogue waves".*

Here the reader may become confused in view of the title. The text lacks a deep description of the theory and the reader will likely be induced to believe having solitons will lead to rogue waves. Reading this statement, it induces a conclusion that solitons don't affect rogue waves. **If you add the main theory/results of Bruhl and Oumeraci (2016) a line before, all will be clear. Solitons are the "fabric" of the irregular wave train and will be there regardless of the tallest waves according to Bruhl and Oumeraci (2016).**

*"The aim of the study was to explore the role of the determined solitons for the generation of rogue waves. In the first part of the study, it was investigated whether specific solitons in the NLFT spectrum could be associated with the recorded rogue waves."*

Here the authors use a very vague set of words. I suspect they meant "individual solitons". **They must be more precise**.

*"For this purpose, all free-surface elevations between the two zero-crossings of a rogue wave (or largest wave, for normal samples) were scaled down to 80 % (Fig. 6)."*

This methodology has no physical, mathematical or statistical reasoning in the manuscript. The authors fail to give a reference. It suggests they implement an arbitrary method to check soliton "associability" with each individual rogue wave. In figure 6c the largest soliton "associated" with the rogue wave did not decrease its amplitude to 80%, rather to $\sim$60%. **Unless the authors provide a strong argument with a solid base for this approach, I recommend to completely remove this artificial change that by no means reflect reality (i.e. the observation). Alternatively, having in hands more than 5 million waves from Norderney, the authors could compare different and yet identical time series but with different wave heights of the rogue waves, and check whether they find the associated soliton. As a second alternative, the authors could run a simulation that creates identical time series with distinct heights for rogue waves**[13]. To highlight the arbitrary nature of this procedure, the authors claim on lines 220-223:

*"The KdV-NLFT was then repeated for the modified time series, which resulted in a new soliton spectrum. It was monitored which of the solitons had changed in amplitude A (and, therefore, in frequency F), due to the change in wave height. These solitons were assumed to have the same position as the rogue/ maximum wave."*

It is not clear why they can assume it is the same position. Although Bruhl and Oumeraci (2016) resonate deterministic reasoning, this above statement shows full arbitrariness. Likewise, the start of section 3.1 discusses how they "associated" solitons with rogue waves through this artificial method:

*"The solitons with constant amplitudes can be regarded not to be associated with the rogue wave."*

Which again demonstrates its lack of analytical and deterministic approach. It is quite surprising, since formation mechanisms dealing with spectral analysis are supposed to tackle deterministic approaches. A further criticism deals with the fact that we have no idea of the depth $kh$ in the discussed examples. **Unless the authors rewrite the text as "(...) amplitudes ARE associated with the rogue wave because (...)", I do not see the relevance of this approach.** In addition, on line 235:

*"Although often the case, the largest soliton attributed to the rogue wave was not necessarily the largest soliton in the spectrum (Fig. 7)."*
* * *
[13]Removing this method/result does not necessarily impact on the overall results. I deem reasonable that the soliton spectrum effect may actually be translated to the mean normalized soliton amplitude. As a third alternative the authors could compute probability curves for mean normalized soliton amplitudes.

Do the authors have any idea of why that is? Again lacks deterministic understanding. **In that case, I see as the only remedy to resort to the statistical understanding.** Although line 250 asserts that the author's results agree with Osborne et al. (1991)[14], the latter study showed that each soliton was uniquely locked with each wave group, as opposed to the current manuscript. So far, the authors speak of a deterministic process but actually talk of statistical uncertainties, like "may" or "may not" be associated for both ordinary and rogue waves. Hence, that is why I believe the authors have plotted figure 8. Unlike the previous figures, figure 8 does contain significant statistical knowledge. **But given the strong ambiguity in the "association" method, I recommend the authors to change the plots for either mean soliton amplitude vs. maximum height or maximum soliton amplitude vs. maximum height, but without any "association".**

In addition, as summarized at the start of this section, the whole association process between individual solitons and individual rogue waves that cover three pages is simply dismissed as a formation mechanism for rogue wave on lines 254-258:

"*The interaction of unidirectional solitons, however, as described by KdV, is known not to result in exceptional increases in wave elevation (Kharif and Pelinovsky, 2003). Hence, the soliton spectrum alone does not yield a satisfactory explanation of the generation mechanism of extreme waves/ crests. One may speculate that the formation of the rogue wave in these cases is a result of the interaction of one or several solitons with the underlying oscillating wave field, a hypothesis which will need further analyses to be validated.*"

The above statement is quite confusing and adds to the lack of organization of the article. If the authors strongly believe the soliton spectrum to show rogue wave formation, this discussion should be place at the end of the paper. On the other hand, if they believe the soliton spectrum to not be able to understand rogue wave formation, **given all the criticism above, I wonder why spend so many pages with an arbitrary approach that can not explain rogue wave formation. I suggest shrinking the whole soliton spectrum development to less than half a page, and give more attention to the requested revised figure 8 that is statistically significant.**

Lines 259-261 repeat the same contradiction of lines 254-258, this is redundant. But more importantly, authors try to remove the influence of the sea state as follows:

"*To remove the influence of the underlying sea state, the soliton amplitudes $A_s^1$ were normalised by the significant wave height $H_s$ of the corresponding sample.*"

Except that normalizing the wave height or soliton height does not remove possible nonlinearities due to the underlying sea state. Proof: We may take the probability density of absolute wave heights for a Gaussian sea (Massel, 2017):

$$f(H) = \frac{H}{4m_0} e^{-H^2/8m_0} \quad , \quad m_0 = \int_0^\infty S(\omega)\, d\omega \quad . \tag{1}$$

The underlying "sea state" in this case causes Gaussianity (Rayleigh distribution), however, by normalizing the height into $H^* = H/H_s$ does not change the Gaussianity:

$$f(H)dH = f(H^*)dH^* \quad \therefore \quad f(H^*) = 4H^* e^{-2H^{*2}} \quad , \tag{2}$$

and thus not changing the underlying sea state. The sea state is not simply defined by $H_s$, and the plot of $f(H/H_s)$ may have sea state peculiarities even in deep water (see Karmpadakis et al. (2020)). Indeed, by plotting the exceedance probability of $H/H_s$ the sea state peculiarities were by no means removed in figures 7-9 of Teutsch et al. (2020). **Therefore, I believe the conclusions based on this statement are misleading as to the removal of the effect of sea states. I recommend authors to plot exceedance probabilities of $H/H_s$ for ranges of $A_s^1/H_s$ and varying sea parameters such as steepness, bandwidth, depth $kh$ or Ursell number. I strongly believe the true influence of solitons will abide in these plots.**
* * *
[14]Please provide the exact quote from this paper that is assured to be in agreement, as I could not find it.

3H. Yet another statement hard to reconcile with the manuscript main line of thought:

*"Brühl (2022)[15] classifies waves with $0.559 \leqslant U$[16] as solitary-like wave types and waves with $U < 0.559$ as Airy-like, Stokes-like or cnoidal-like. For our data, in which the bulk of waves are located below $U = 0.559$, this means that most rogue wave crests are not soliton-like."*

While Bruhl and Oumeraci (2016) assured that all periodic oscillatory waves in the range of validity of the KdV equation are composed of solitons interacting with each other, what does this above statement mean? How can these waves be of no soliton-like structure and yet be composed of solitons? **Please reconcile both statements in a clear fashion, if possible.** To add even more confusion, the authors portray no surprise, and claim that it is expected the rogue waves can not be explained by solitons alone on line 294. They also add that maybe solitons have to interact with other wave components, but they have not analyzed it. **If none of the above is reconcilable, I recommend the authors to include the literature discussion that precludes rogue waves from being formed by solitons alone in the first pages of the manuscript, and move on directly to what can be attained. It requires the reader many pages to understand the relevance of a soliton spectrum, only to learn that it alone can not achieve the expected purpose.** Then figure 10 discusses soliton amplitudes versus Ursell number, and authors observe a saturation in the growth of the former until on lines 298-303 admit that there is no explanation for this phenomenon. **Then, I ask: what is the relevance of figure 10? It would be better to plot exceedance probability for varying Ursell and fixed soliton amplitudes.**

3I. After a series of contradictions in 3H, The authors jump to the next section (3.2) seemingly assuring that soliton spectrum can reveal formation of rogue waves, just after the paragraph that said the latter task to be impossible!

*"When investigating the attribution of solitons to rogue waves in Sect. 3.1, we found in the majority of cases that the largest soliton in the nonlinear spectrum could be attributed to the rogue wave. In addition, this soliton was often outstanding from the other solitons in the spectrum, with a much larger amplitude than the remaining solitons in the spectrum (see the example in Fig. 6). We were therefore interested in whether the existence of an outstanding soliton in the nonlinear spectrum was typical for rogue wave samples off Norderney. We investigated this question statistically by comparing soliton spectra, calculated from 310 vKdV-NLFT, for normal samples and the four different categories of rogue wave samples".*

The comments extracted from lines 294-303 on page 21 of the manuscript clearly dismisses what is written on page 22. **I do not understand why the text insists on soliton spectrum after being dismissed at least three times earlier. If the authors want to discuss the statistical inference from soliton spectrum without damaging the reader's ability to understand and without challenging the line of thought on every page, I recommend that the caveats and issues of the soliton spectrum be either transferred to the introduction or the discussion as a counterpoint.**

3J. On page 23 leading to equation 13 the authors implement yet another arbitrary tool[17] without any solid justification nor references. Nevertheless, even so the results in table 2 demonstrate large percentage of false positives for "normal" samples (which surely include many relevant waves in the range $1.5 < H/H_s < 2$) and for "height" samples. The authors correctly conclude that outstanding solitons (whatever arbitrary notion defined them) can not be a strong predictor[18] for height rogue waves or large ordinary waves near the rogue threshold, and hence, it is a good predictor only for extreme rogue waves. **The problem here is that this statement is not found in the conclusions. Please add this discussion to the Conclusions.** I would expect something of the type:
* * *
[15] Attention to this reference that can not be found.

[16] Why don't you write $U \geqslant 0.559$?

[17] I wonder if the variance of the soliton spectrum would not be a better tool.

[18] Here predictor does not mean it has predictive power, as one can not extract the soliton spectrum before the time series materializes.

The literature shows that solitons alone can not form rogue waves, and it is hypothesized that when the former interact with linear waves the latter can be formed. Nevertheless, we carried out a clustering procedure to separate normal and outstanding solitons loosely associated with rogue waves, and found that only for extreme rogue waves the soliton spectrum analysis is a good predictor.

3K. At the start of the conclusions it is stated:

*"Rogue wave occurrence recorded off the coast of the island Norderney is not sufficiently explained by second-order theory."*

Where in Teutsch et al. (2020) have second-order models in steepness been evaluated? You mean Weibull model, no? **Please be careful with your jargon. You could test an actual second-order model for wave crests, like Forristall (2000).** This text in the conclusions point to the fact that indeed in 1B (see page 1) they were describing Forristall (1978) as a second-order model, a clear mistake.

3L. In the concluding remarks I find:

*"Rogue waves at Norderney are likely to be a result of the interaction of solitons with the underlying field of oscillatory waves."*

This is a big leap not supported by the manuscript, considering that you did not study this type of interactions (see line 258). **I believe the authors can not discuss likelihood of a process that was not analyzed. I recommend you change "are likely" to "Despite not assessing the interaction of solitons with oscillatory waves, we are confident that rogue waves at Norderney could be a result...".**

**Conclusion**
* * *
The reviewer thanks for the opportunity to read this important work. Overall, I support the publication of this preprint once all these issues have been clarified and/or amended/removed.

**References**
* * *
Baschek, B., Imai, J., 2011. Rogue wave observations off the us west coast. Oceanography 24, 158 – 165.

Benjamin, T.B., Feir, J.E., 1967. The disintegration of wave trains on deep water part 1. theory. Journal of Fluid Mechanics 27, 417–430.

Bitner-Gregersen, E., Gramstad, O., 2015. Rogue waves: Impact on ships and offshore structures. DNV GL Strategic Research & Innovation Position Paper .

Bolles, C., Speer, K., Moore, M., 2019. Anomalous wave statistics induced by abrupt depth change. Physical Review Fluids 4.

Bruhl, M., Oumeraci, H., 2016. Analysis of long-period cosine-wave dispersion in very shallow water using nonlinear fourier transform based on kdv equation. Applied Ocean Research 61, 81–91.

Cartwright, D., Longuet-Higgins, M., 1956. The statistical distribution of the maxima of a random function. Proc. R. Soc. A 237, 212–232.

Cattrell, A., Srokosz, M., Moat, B., Marsh, R., 2018. Can rogue waves be predicted using characteristic wave parameters? J. Geophys. Res. Oceans 123, 5624–5636.

Dean, R., Dalrymple, R., 1984. Water wave mechanics for engineers and scientists. World Scientific .

Dingemans, M.W., 1997. Water Wave Propagation Over Uneven Bottoms. World Scientific.

Doleman, M.W., 2021. Rogue waves in the dutch north sea. Master's thesis, TU Delft .

Draper, L., 1964. Freak ocean waves. Oceanus 10, 13–15.

Draper, L., 1971. Severe wave conditions at sea. J. Inst. Navig. 24, 273–277.

Earle, M.D., 1975. Extreme wave conditions during hurricane camille. J. Geophys. Res. 80, 377–379.

Fedele, F., Brennan, J., De Leon, S., Dudley, J., Dias, F., 2016. Real world ocean rogue waves explained without the modulational instability. Sci. Rep. 6, 27715.

Forristall, G., 1978. On the distributions of wave heights in a storm. J. Geophys. Res. 83, 2353–2358.

Forristall, G., 2000. Wave crest distributions: observations and second order theory. J. Phys. Ocean. 30, 1931–1943.

Fu, R., Ma, Y., Dong, G., Perlin, M., 2021. A wavelet-based wave group detector and predictor of extreme events over unidirectional sloping bathymetry. Ocean Eng. 229.

Glukhovskii, B., 1966. Investigation of sea wind waves (in russian). Gidrometeoizdat .

Gramstad, O., Zeng, H., Trulsen, K., Pedersen, G., 2013. Freak waves in weakly nonlinear unidirectional wave trains over a sloping bottom in shallow water. Physics of Fluids 25.

Häfner, D., Gemmrich, J., Jochum, M., 2021. Real-world rogue wave probabilities. Scientific Reports 11.

Haring, R., Osborne, A., Spencer, L., 1976. Extreme wave parameters based on continental shelf storm wave records. Proc. 15th Int. Conf. on Coastal Engineering, Honolulu, HI , 151–170.

Huntley, D., Guza, R., Bowen, A., 1977. A universal form for shoreline run-up spectra? Journal of Geophysical Research 82, 2577–2581.

Jahns, H., Wheeler, J., 1973. Long-term wave probabilities based on hindcasting of severe storms. J. Petrol. Technol. 25, 473–486.

Jorde, S., 2018. Kinematiken i bølger over en grunne. Master's thesis, University of Oslo .

Karmpadakis, I., Swan, C., Christou, M., 2020. Assessment of wave height distributions using an extensive field database. Coastal Eng. 157.

Kharif, C., Pelinovsky, E., 2003. Physical mechanisms of the rogue wave formation. Eur. J. Mech. B Fluids 22, 603–634.

Lawrence, C., Trulsen, K., Gramstad, O., 2021. Statistical properties of wave kinematics in long-crested irregular waves propagating over non-uniform bathymetry. Physics of Fluids 33.

Longuet-Higgins, M., 1952. On the statistical distribution of the heights of sea waves. Journal of Marine Research 11, 245–265.

Longuet-Higgins, M., 1963. The effect of non-linearities on statistical distributions in the theory of sea waves. J. Fluid Mech. 17, 459–480.

Longuet-Higgins, M.S., 1975. On the joint distribution of the periods and amplitudes of sea waves. J. Geophys. Res. 80, 2688–2694.

Mallory, J., 1974. Abnormal waves in the south-east coast of south africa. Int. Hydrog. Rev. 51, 89–129.

Massel, S., 2017. Ocean surface waves: Their physics and prediction. 3rd ed., World Scientific, Singapore.

Mendes, S., Scotti, A., 2021. The rayleigh-haring-tayfun distribution of wave heights in deep water. Applied Ocean Research 113, 102739.

Mendes, S., Scotti, A., Brunetti, M., Kasparian, J., 2022. Non-homogeneous model of rogue wave probability evolution over a shoal. Accepted at J. Fluid Mech. ; Preprint at EarthArXiv doi:https://doi.org/10.31223/X5NG85.

Olagnon, M., van Iseghem, S., 2000. Some cases of observed rogue waves ad an attempt to characterize their occurrence conditions. In: M. Olagnon and G.A. Athanassoulis (Eds.), Rogue Waves 2000 , 105–116.

Orzech, M.D., Wang, D., 2020. Measured rogue waves and their environment. Journal of Marine Science and Engineering 8.

Osborne, A.R., Segre, E., Boffetta, G., Cavaleri, L., 1991. Soliton basis states in shallow-water ocean surface waves. Phys. Rev. Lett. 67, 592–595.

Raustøl, A., 2014. Freake bølger over variabelt dyp. Master's thesis, University of Oslo .

Rice, S., 1945. Mathematical analysis of random noise. Bell Syst. Tech. J. 24, 46–156.

Slunyaev, A., Sergeeva, A., Didenkulova, I., 2016. Rogue events in spatiotemporal numerical simulations of unidirectional waves in basins of different depth. Natural Hazards 84, 549 – 565.

Stansell, P., 2004. Distribution of freak wave heights measured in the north sea. Appl. Ocean Res. 26, 35–48.

Tayfun, M.A., Fedele, F., 2007. Wave-height distributions and nonlinear effects. Ocean Eng. 34, 1631 – 1649.

Teutsch, I., Weisse, R., Moeller, J., Krueger, O., 2020. A statistical analysis of rogue waves in the southern north sea. Natural Hazards and Earth System Sciences 20, 2665–2680.

Trulsen, K., Raustøl, A., Jorde, S., Rye, L., 2020. Extreme wave statistics of long-crested irregular waves over a shoal. J. Fluid Mech. 882.

Wu, Y., Randell, D., Christou, M., Ewans, K., Jonathan, P., 2016. On the distribution of wave height in shallow water. Coastal Eng. 111, 39–49.

Xu, J., Liu, S., Li, J., Jia, W., 2021. Experimental study of wave height, crest, and trough distributions of directional irregular waves on a slope. Ocean Engineering 242, 110136.

Zhang, J., Benoit, M., 2021. Wave–bottom interaction and extreme wave statistics due to shoaling and de-shoaling of irregular long-crested wave trains over steep seabed changes. Journal of Fluid Mechanics 912, A28.

Zheng, Y., Lin, Z., Li, Y., Adcock, T., Li, Y., Van Den Bremer, T., 2020. Fully nonlinear simulations of unidirectional extreme waves provoked by strong depth transitions: The effect of slope. Phys. Rev. Fluids 5.

Zou, L., Wang, A., Wang, Z., Pei, Y., Liu, X., 2019. Experimental study of freak waves due to three-dimensional island terrain in random wave. Acta Oceanologica Sinica 38, 92–99.

---

## Referee Comment (RC2)

**Review**
on the manuscript "*Contribution of solitons to enhanced rogue wave occurrence in shallow water:
a case study in the southern North Sea*" by I. Teutsch, M. Bruhl, R. Weisse, S. Wahls
submitted for publication in the **NHESS** journal

In this paper the in-situ data are analyzed using the Inverse Scattering Transform (IST) under the assumption that the waves may be approximated by the Korteweg – de Vries equation. An abnormal statistics of waves measured in a particular site in shallow waters off the coast of the island Norderney, motivated the research. In contrast to measurements at other locations, analyzed in the preceding paper by Teutsch et al. (2020), measurement by a buoy at Norderney demonstrated an enhanced rogue wave occurrence probability, which could not be explained by the Forristall distribution. To the best of my knowledge this is the first time when in-situ measurements are analyzed systematically with respect to the soliton content. The authors show a clear correlation between rogue wave occurrences and the presence of 'outstanding' solitons in the wave fields, and do their best to prove reliability of this result.

It is generally accepted that the 'nonlinear spectrum' provided by the IST should better characterize the dynamical properties of waves with prominent nonlinear coherence. Therefore I appreciate the approach and the work in general. However, the manuscript in its present form has two serious drawbacks. First, the introductory part is written unacceptably badly. Its author is obviously not a specialist in the topic of integrable equations and related issues. The introduction contains statements which are wrong, illogical or can be easily misinterpreted; the references are not always appropriate. There are too many issues to be listed all in my report. I assume that this problem may be solved with the help of co-authors of the work, some of whom seem to be highly qualified in the concerned field of science.

The other problem is that the analysis is based on the assumption that waves may be approximated by the KdV equation, and the revealed by the IST discrete spectrum indeed corresponds to physical solitons which are long-lived wave structures which may be measured upstream and downstream the location of measurements. This point cannot be justified on the basis of single-point measurements. From the general viewpoint, fields of KdV solitons are typically characterized by very asymmetric (cnoidal-type) waves, so that solitons could be recognized by eye. Nothing of this kind can be seen in the presented waveforms. Besides, the estimated soliton amplitudes are not very large, the solitons do not dominate. My personal opinion is that the chance, that the conclusion of the work is wrong, is 80%. The revealed solitons most likely characterize the wave shapes, which do not correspond to physical long-lived KdV-like solitons. I express my alternative vision below. I assume that the manuscript may be published only with a clear discussion that the employed assumption is crucial, but cannot be firmly justified within the frames of the available data. This should be mentioned in the abstract and the conclusion as well.

I have two alternative ideas why the enhanced probability of large waves is observed at this site. First, the location of measurements corresponds to a strongly varying bathymetry (line 138). The authors mention the recent progress in understanding the underlying mechanisms of rogue waves occurring under the depth change conditions (line 376). I wish to draw the attention to another recent work [Ducrozet et al, 2021], where envelope solitons are shown responsible for the wave amplification when the depth increases.

The second idea is based on the discussion in lines 108 and further on. There the theoretical result, which describes four-time wave amplification due to the oblique interaction of KdV solitons, is mentioned. It is an essentially directional effect, which indeed may explain peculiarities of the statistics for crossed shallow water waves. This is somehow confirmed in lines 407-408. However, this effect cannot be considered within the KdV theory, and hence it cannot be revealed using the KdV-NLFT. If this effect, however, takes place indeed, the amplified waves may be interpreted by the KdV-NLFT as ones containing intense solitons, while this will not correspond to the physical essence of the phenomenon.

See also my comment to lines 276-277.

I wish to draw the authors' attention to the work [Slunyaev et al, 2006] where NLS solitons revealed in the instrumental records of deep-water waves using the IST were related to rogue wave events. Accuracy of this procedure for strongly nonlinear waves was estimated in [Slunyaev, 2018]. Observation of a long-lived NLS soliton in the field of strongly nonlinear waves was made in the numerical simulations by Slunyaev (2021) and in-situ in [Onorato et al, 2021].

The KdV-soliton content in sinusoids was discussed in the work by Giovanangeli et al (2018); it may be useful.

Lines 44, 83. Application of the NLS theory is not limited by the condition $kh > 1.363$. The de-focusing NLSE may be derived for shallow water waves. The focusing (deep water) and defocusing (shallow water) NLSEs are equations on wave modulations, while the KdV equation describes the wave displacement. Therefore the phrase "*The shallow-water equivalent to the NLS equation is the Korteweg–de Vries*" in line 83 is essentially inaccurate.

Line 48. Under the deep-water condition, the most unstable perturbations are directed in the longitudinal direction, not oblique, see [McLean, 1982a,b].

Line 52. I believe that the BFI parameter was introduced for characterization of irregular wave statistics for the first time in [Onorato et al, 2001].

Line 55. I would not say that exact breather solutions explain the physics of the modulational instability. They simply describe the dynamics.

Line 61. The paper by Shrira & Geogjaev was published in 2010, not 2009.

Lines 67-73. It is reasonable to say more precisely, what is called shallow water, in terms of the dimensionless depth. At $kh < 0.5$ the effect of the modulational instability disappears.

Line 78. Below in the text, the authors mention a great number of recent works on the effect of rogue waves under the conditions of variable depth. Hence, the statement "*so far only few studies have addressed the impact of bathymetry on rogue wave generation*" is not consistent. The effect of finite depth on the structure of rogue waves described by the NLSE was directly discussed in [Didenkulova et al, 2013].

Lines 97-99. The works by Zabusky & Kruskal (1965) and Peregrine (1983) are fundamental in this story, but the references to them are inappropriate. In the former paper they consider periodical domain, hence could not observe the wave decay; while the latter paper is dedicated to the NLSE, not KdV. It is better to cite here some classical textbook on the IST in infinite line.

Line 102. The result of the work by Pelinovsky et al. (2000) is inverted. In fact, they show that the wave train contains **at most** one soliton, not at least.

Lines 103, 257. The guess that the interaction between many KdV soliton can lead to a rogue wave formation is not supported by the theoretical findings [Slunyaev, 2019] and direct numerical simulations, e.g. [Dutykh & Pelinovsky, 2014; Pelinovsky & Shurgalina, 2017]. Thus, it is wrong.

Line 117. There are two more recent publications from this series: Costa et al. (2014) and Osborne et al. (2019).

Fig. 2. I suggest to show the scale (meters) in the map.

Eq. 1. Why this threshold for the shallow-water condition is used? It is related to the Benjamin – Feir instability, but not to the applicability of the KdV theory…

Line 155. I believe, the number '1' should be removed from the formula for $T_p$.

Table 1. The caption is wrong.

Table 1. In the time series the data acquisition frequency is rather low, thus the waveform resolution is poor, what may complicate analysis of the wave shapes. According to the table, the amount of

rogue waves of the types "crest rogue", "double rogue" and "extreme rogue" are at most 1%. How reliable is the casting into these classes?...

Line 211. I am not sure that the words "*the frequency axis has no physical meaning*" are correct. As I understand, the horizontal axis in fact corresponds to the inverse duration of the soliton. Since for the KdV solitons the relation $AT^2$ = Const holds (where $A$ is the soliton amplitude and $T$ is its duration, see Eq. (12)), then the "frequency" (~1/T) may be a more sensitive parameter than the soliton amplitude. This property is used de-facto by the authors to distinguish the "outstanding" solitons.

Lines 267, 269. The words "right" probably do not correspond to Fig. 9.

Lines 276, 277. I interpret the conclusion in the way that outstanding solitons are not indicators of rogue waves (in terms of the quantity $H/H_s$ or similar), but are indicators of large waves in general. Then, I can assume that they do not correspond to physical solitons, but rather artifacts of application of the weakly nonlinear theory to the analysis of strongly nonlinear waves. Hence, they are fakes.

A similar conclusion seems to follow from lines 428-429, where it is mentioned that "double" and "extreme" rogue samples exhibit the effect most clearly.

Fig. 10. The dimensions of the vertical axes are not given.

Table 3. I suggest to mention in the discussion of this table, that depending on the depth, the amplitudes of the solitons change. The soliton amplitude is controlled by the square root of the Ursell parameter (12), $U \sim h^{-3}$, hence the amplitudes should depend on depth as $h^{-3/2}$. Therefore, a 20% change of the depth will result in roughly 30% change of the amplitudes, if my estimations are correct.

References

Costa, A., Osborne, A.R., Resio, D.T., Alessio, S., Chrivi, E., Saggese, E., Bellomo, K., Long, C.K. (2014) Soliton Turbulence in Shallow Water Ocean Surface Waves. Phys. Rev. Lett. 113, 108501.

Didenkulova, I.I., Nikolkina, I.F., Pelinovsky, E.N. (2013) Rogue waves in the basin of intermediate depth and the possibility of their formation due to the modulational instability. JETP letters 97, 194-198.

Ducrozet, G., Slunyaev, A.V., Stepanyants, Y.A. (2021) Transformation of envelope solitons on a bottom step. Physics of Fluids 33, 066606.

Dutykh, D., Pelinovsky, E. (2014) Numerical simulation of a solitonic gas in KdV and KdV–BBM equations. Phys. Lett. A. 378, 3102-3110.

Giovanangeli, J.-P., Kharif, C., Stepanyants, Y.A. (2018) Soliton spectra of random water waves in shallow basins. Math. Model. Nat. Phenom. 13, 40.

McLean J.W. (1982b), Instabilities of finite-amplitude water waves. J. Fluid Mech. 114, 315-330.

McLean, J.W. (1982a) Instabilities of finite-amplitude gravity waves on water of finite depth. J. Fluid Mech. 114, 331-341.

Onorato, M., Cavaleri, L., Randoux, S., et al. (2021) Observation of a giant nonlinear wave-packet on the surface of the ocean. Sci. Rep. 11, 23606.

Onorato, M., Osborne, A.R., Serio, M., Bertone, S. (2001) Freak waves in random oceanic sea states. Phys. Rev. Lett. 86, 5831–5834.

Osborne, A.R., Resio, D.T., Costa, A., et al. (2019) Highly nonlinear wind waves in Currituck Sound: dense breather turbulence in random ocean waves. Ocean Dynamics 69, 187–219.

Pelinovsky, E., Shurgalina, E. (2017) KDV soliton gas: interactions and turbulence. In.: Advances in Dynamics, Patterns, Cognition: Challenges in Complexity (Nonlinear Systems and Complexity, 20) (Eds.: I. Aronson, A. Pikovsky, N. Rulkov, L. Tsimring), Vol. 20, 295−306.

Slunyaev, A. (2006) Nonlinear analysis and simulations of measured freak wave time series. European J. of Mechanics B / Fluids 25, 621-635.

Slunyaev, A. (2019) On the optimal focusing of solitons and breathers in long wave models. Studies in Applied Mathematics 142, 385-413.

Slunyaev, A.V. (2018) Analysis of the nonlinear spectrum of intense sea waves with the purpose of extreme wave prediction. Radiophysics and quantum electronics 61, 1-21.

Slunyaev, A.V. (2021) Persistence of hydrodynamic envelope solitons: detection and rogue wave occurrence. Physics of Fluids 33, 036606.

---

## Author Comment (AC1)

**Reply to review #1**

We thank Referee #1 for the constructive comments that have helped us to clarify and improve many aspects of our manuscript. In the following, we explain how we plan to address the individual issues raised by the reviewer in the revised manuscript.

**1. General Comments**

**1A** We thank the reviewer for this comment. We will add a proper definition of rogue waves and discuss their relevance, as suggested.

**1B** We thank the reviewer for pointing out this ambiguity in the manuscript. We will draw a clear connection of Rayleigh and Weibull distributions to linear superposition and add an additional reference for second-order models to the text.

**1C** We appreciate the critical review of the paragraph. We agree that some formulations are misleading and we will re-formulate the passage in agreement with the suggestions of the reviewer.

**1D** The word "respective" will be removed, as suggested by the reviewer.

**1E** We thank the reviewer for pointing out additional important conclusions in the literature. We will add these to our introduction.

**1F** We agree with the reviewer. We will add the suggested references accordingly.

**1G** We agree with the reviewer. We will add the suggested references accordingly.

**1H** We understand the objection of the reviewer. In our text, we will replace the expression "nonlinearity" by "nonlinear processes", where appropriate.

**1I** We thank the reviewer for the additional reference. We will add it accordingly.

**1J** We agree with the reviewer on the contradiction in the text. We will rephrase the sentence in question accordingly.

**1K** We thank the reviewer for several additional important references concerning rogue wave occurrence on varying bathymetry. We will extend the discussion and include the suggested references.

**1L** We thank the reviewer for this remark. We will replace the unpublished reference by publically available literature.

**2. Scientific Improvement**

**2A** We thank the reviewer for this detailed assessment and the comparison with the previous study of Teutsch et al. (2020). The reviewer is completely right that Figures 7 and 9 in Teutsch et al. (2020) do not support the conclusion that rogue wave frequencies are overestimated by the Forristall distribution. However, in these Figures data from different types of instruments (radar and wave buoys) are considered jointly. Figure 2 in Teutsch et al. (2020) clearly shows that data from both instruments show different behaviour with rogue wave frequencies in the radar/buoy data set being higher/lower than that derived from the Forristall distribution. The only exception here were the results from the buoy SEE off Norderney, which showed results comparable to those derived from radar data. This rendered the station SEE outstanding and provided the motivation for this study.

**2B** We thank the reviewer for his/her discussion of the Ursell number formulation. As suggested, we will rewrite the equation and state in the text that different definitions of the Ursell number exist, which will lead to different threshold values.

**2C** For the agreement between the Forristall distribution and the measurements, please see our reply to comment 2A. In l. 369 of the discussion, we will add the information that the "nearby stations" were buoy stations as well, to avoid confusion of buoy and radar measurement results from the previous study.
Regarding the second part of the comment, we acknowledge that there is a clear definition of what represents shallow/deep/intermediate water for a wave, while the terms were used here in a broader sense to distinguish sites. We will revise the manuscript to make this clearer and we will use a different terminology. We will also replace second-order theory by Forristall distribution.

**2D** The mentioned citation refers to non-Gaussianity in decreasing water depth, here in the context of wave run-up. The article is referred to by Sergeeva et al. (2011) to emphasise nonlinear behaviour of waves above a varying bathymetry. We do, however, agree with the reviewer that our formulation concerning the reference is misleading, as the referred article does not concern rogue waves. We will therefore re-formulate the sentence to read "... described e.g. by Huntley et al. (1977) *in the context of wave run-up. It has* gained increased attention in the context of rogue wave occurrence *(e.g. Sergeeva et al.* (2011))".

**3. Scientific Issues**

**3A** We thank the reviewer for raising the issue of the shallow water definition. We will present the ranges of $kh$ in our data, as well as the value of the slope, as suggested. The reviewer is right that our article does not solely concern waves in shallow water as defined by $h < L/20$. Since we investigate waves in the context of the KdV equation, we follow the definition of the applicability of the KdV equation as given by Osborne and Petti (1994), p. 1731, and Osborne (1995), p. 2629. We acknowledge that, therefore, our definitions of shallow water ($kh < 1.36$ or $h/L < 0.22$) and deep water ($kh > 1.36$ or $h/L > 0.22$) are different from the definitions of shallow ($h/L < 0.05$), intermediate ($0.05 < h/L < 0.5$) and deep water ($h/L > 0.5$) that are used in the engineering context, and that this may lead to confusion. We will therefore state this difference more clearly in the text, and we will clearly define the

terminology for 'shallow' that is used in the paper. Furthermore, we will change the title to include "shallow depths" instead of "shallow water", as suggested by the reviewer.

**3B** We apply the definition of the KdV equation as given in Osborne (2010) , p. 9, which defines the linear phase speed as $c_0 = \sqrt{gh}$. We agree that the term 'shallow-water wave celerity' is misleading when applying the KdV equation to relative depths larger than $h/L$ = 0.05. Nevertheless, the linear phase speed is used in the KdV equation within the range of applicability. We will use the term 'linear phase speed' instead.

**3C** We thank the reviewer for pointing out that the category "normal" does not account for the fact that waves slightly below the threshold $H/H_s$ = 2.0 are influenced by nonlinear processes and can become highly dangerous, similarly to waves with $H/H_s$ > 2.0. We will change the category "normal" to "non-rogue", to emphasise that these samples do not include waves according to the definitions $H/H_s$ ≥ 2.0 or $C/Hs$ ≥ 1.25.

**3D** We agree with the reviewer that the number of samples is a vague quantity for the reader and that the total number of waves should be more informative. We will include the precise number of measured waves in Table 1.

**3E** We thank the reviewer for the valuable suggestion that the results from the crucial papers Osborne et al. (1991) and Bruehl and Oumeraci (2016) should be explained to the reader more thoroughly. For explanation, we will additionally refer to earlier crucial references, the original numerical studies by Zabusky and Kruskal (1965) and Osborne & Bergamasco (1986). Osborne et al. (1991) applied the approach to ocean measurement data and Bruehl and Oumeraci (2016) performed an experimental study. Note that there is a technical difference between our approach and the approach in the three last mentioned works. We use the NLFT for vanishing boundary conditions, while Osborne & Bergamosco (1986), Osborne et al. (1991) and Bruehl and Oumeraci (2016) apply the NLFT for periodic boundary conditions. The relation between the two transforms is somehow similar to the relation between the linear Fourier transform and the linear Fourier series (which is what the FFT computes). They are related, and both have their advantages and disadvantages, but not all results can be directly compared for this reason. We will add paragraphs accordingly, additionally referring to earlier crucial references that explain the behaviour of solitons for the NLFT employed in our paper, like Hammack and Segur (1974) and Ablowitz & Kodama (1982). In our context, the sea surface elevation is described by a discrete spectrum indicating solitons and a continuous spectrum indicating a dispersive wave train. Of these two parts, we only discuss the soliton spectrum further in this article. However, it is known that for vanishing boundary conditions the soliton spectrum completely describes the behaviour of the wave train in the far field. After the complete dissipation of the dispersive waves, only the solitons are left in the far field. When the distance between these solitons is sufficiently large, no interactions occur between them and all solitons are clearly visible with their characteristic shapes. Assuming frictionless propagation, their amplitudes can already be read from the nonlinear spectrum of the initial time series. Therefore, we prefer to add the equation for the surface elevation in the far field, resulting from the solitons, given e.g. by Equation (4) in Prins & Wahls (2019), with reference to Schuur (1984), Eq. 17, Schuur (1986), p. 83, Eq. 33 and Ablowitz & Kodama (1982), Eq. 2.20a.

We further agree to add plots that explain the meaning of the soliton spectrum (Figure 1 in this response). In addition to an exemplary time series (blue line in the first plot, zoomed in to the rogue wave), we will add its linear FFT spectrum (second plot) and, in addition to the soliton spectrum (last plot), the nonlinear continuous spectrum (third plot, which will not be analysed further in this article). Each of the solitons in the soliton spectrum would be a physical soliton if the signal is propagated according to the KdV equation. After sufficiently long propagation, each of these solitons will appear isolated with its characteristic shape. Within the time series, the solitons are close together; they overtake and interact with each other. For visualisation of the role of the solitons in the time series, the first plot shows the soliton train (red line) that is obtained by nonlinear superposition of the solitons (considering their interactions) using the algorithm from Prins & Wahls (2021). Note that inverting large soliton spectra is numerically very difficult (Prins & Wahls, 2021). We therefore had to use a shortened time series for the figure.

[Figure]

*Figure 1: a) time series and soliton train, as calculated from the inverted soliton spectrum. b) linear Fourier spectrum. c) continuous spectrum. d) discrete soliton spectrum.*

**3F** The reviewer is right in that the NLFT is currently employed as a trace method. The reason is that we do not know how the nonlinear spectrum changes during propagation around the buoy. If the KdV equation describes the propagation around the buoy reasonably well, then the nonlinear spectrum would be approximately constant during propagation, and the method could single out certain time series that lead (or led) to extreme rogue waves. This aspect however requires more research that is beyond the scope of the current paper. Therefore, we use the approach as a trace method, which has the additional potential to provide further information in future applications. Our work is nevertheless a necessary step in the direction of recognising potentially dangerous time series. If the method does not work for visible rogue waves, there is little hope for it to work for hidden rogue waves. We demonstrate for the first time that certain distinctive patterns in the nonlinear spectrum of real-world time series indicate extreme rogue waves (at a specific measurement site). Finally, we note that even if the KdV does not describe propagation well, the NLFT could still be a better transform to analyse data in this area than the linear Fourier transform (where the spectrum also only develops in a simple way if the propagation is linear and the depth is constant, but it is nevertheless applied in different contexts). We will discuss these points in the revised manuscript.

**3G**
- As suggested, we will support the explanation of the results by describing the insights from Bruehl and Oumeraci (2016) and the relationship between the soliton spectrum and the far field behaviour under KdV, in a meaningful connection with our reply to 3E.
- The use of the term "determined" was supposed to imply that the soliton was determined by the use of NLFT. For clarity, we will replace the terms "determined" and "specific" with "individual".
- The reviewer criticises that scaling down a rogue wave to 80% is not linked to any physical explanation. We would like to reply that an established method for the treatment of NLFT spectra does not exist. In contrast to the linear case, where the impact of a window on the spectrum can be expressed analytically in a way that is easy to interpret, no such result is known for the nonlinear case. Windowing of the time series and calculating separate nonlinear spectra is common, but does not have any theoretical grounding. In contrast to windowing, which is a general purpose technique that impacts large parts of the time series, our method is local and aims specifically at rogue waves. By scaling only the rogue wave, the changes in the time series are as small as possible. The hope is thus that the danger of evoking additional, unrelated changes in the nonlinear spectrum is minimised by this approach.
  We intend to localise the influence of a change in rogue wave height in the soliton spectrum to establish a connection between a measured rogue wave and individual solitons. The underlying idea of the method is that if local changes of the rogue wave lead to local changes in the spectrum, the changing soliton components are associated with the rogue wave. Since a rogue wave is a particular wave event, it is reasonable to explore changes in the spectrum when only this wave is changed. Furthermore, also the (hidden) solitons in the data are localised components and changes to the particular rogue-wave event are expected to have effects to the soliton spectrum only when a soliton is located sufficiently close to the modified region. The changes in the soliton spectrum only affect a few solitons, whereas all other solitons remain constant. Since only a few solitons are modified, we can conclude that these solitons are located in the modified rogue-wave region within the time series. Regarding the request to remove the rogue wave from the time series, we prefer reducing its height as opposed to cutting it out, which would

introduce an artificial gap to the time series. The method shows that gradually reducing the height of the rogue waves leads to the gradual reduction of individual solitons. A change of the rogue wave will not have an impact on all soliton components in the spectrum. By this straightforward approach, solitons that are directly linked to the rogue wave are easily identified.

- Solitons linked to the rogue wave are not always the largest solitons in the spectrum. As also pointed out in the manuscript, the soliton alone is not sufficient to explain the rogue wave. Only by interaction with components from the continuous spectrum, the rogue wave is formed. In order for this to happen, the soliton and the other components must interact constructively. When the interactions are not constructive, it is very well possible that a larger soliton leads to a smaller hump in the time series. Hence, the dispersive waves and nonlinear interactions have a strong impact, and the largest soliton is not necessarily associated with the largest wave in the time series. For a visual illustration, we would like to refer to Figure 1 (a) in Osborne et al. (1991), in which the largest soliton is also not associated with the highest wave in the time series. In contrast, in our example given above, the largest soliton is located close to the position of the rogue wave.

- Against this background, changing Figure 8, in which rogue wave heights are compared with the associated solitons, would not make sense. Comparing with the highest soliton in the spectrum would include some solitons that are linked to wave groups without rogue waves. While we do not plan to change this in the paper, we nevertheless followed the suggestion of the reviewer and calculated $A_{max}$ with respect to $H_{max}$. We present the updated Figure 8 in comparison with the original Figure 8 below (Figure 2-Figure 4). Here, grey dots show cases, in which the highest attributed soliton is identical with the maximum soliton in the discrete spectrum ($A_s{}^1 = A_{max}$). For rogue samples, this is true in most cases (extreme: 87%, double: 85%, crest 78%, height 71%). For non-rogue samples, this is true in 42% of the cases. The figures show that the results are in a comparable range when rogue wave (or maximum wave) heights are related to the maximum instead of the highest attributed soliton.

[Figure]

*Figure 2: Update of Figure 8a in the preprint. Blue, yellow and green dots, in the legends referred to as "Height", "Double" and "Extreme", represent values from the original figures. When using maximum soliton amplitudes instead of the amplitudes of the attributed solitons, these values change to values represented by red markers. Grey markers show values of attributed soliton amplitude that are identical with the maximum soliton amplitude.*

[Figure]

*Figure 3: Update of Figure 8b in the preprint. Yellow and orange dots, in the legends referred to as "Double" and "Crest", represent values from the original figures. When using maximum soliton amplitudes instead of the amplitudes of the attributed solitons, these values change to values represented by red markers. Grey markers show values of attributed soliton amplitude that are identical with the maximum soliton amplitude.*

[Figure]

*Figure 4: Update of Figures 8c and 8d in the preprint. Dark red dots, in the legends referred to as "Normal", represent values from the original figures. When using maximum soliton amplitudes instead of the amplitudes of the attributed solitons, these values change to values represented by red markers. Grey markers show values of attributed soliton amplitude that are identical with the maximum soliton amplitude.*

- In our study, soliton amplitudes were always smaller than rogue wave crests, which is in agreement with Figure 1a in Osborne et al. (1991).
- We understand that the formulation "To remove the influence of the underlying sea state" in line 261 is misleading. We agree with the reviewer that the influence of the sea state is not only characterised by the significant wave height. Our intention with normalising by $H_s$ is to create dimensionless values, to be able to compare different samples. Since rogue waves are defined on the basis of the significant wave height, we find this parameter suitable for the normalization. The influence of the sea state in terms of the parameters suggested by the reviewer (steepness, Ursell number, $kh$, bandwidth) affect the wave components in the continuous part of the nonlinear spectrum, which we do not discuss further in this article. The soliton spectrum is not affected. Through the continuous spectrum, the sea state parameters possibly influence the wave distribution. We have tested the method that was suggested by the reviewer. As an example, Figure 5 shows the exceedance probability of $H/H_s$ in all samples of a defined Ursell number range. It is seen that the distribution behaves differently in the different ranges. However, this cannot be stated for certain, as the results rely on few data, due to the binning into ranges. The few rogue waves in the samples are distributed randomly, which leads to uncertain results. Together with the consideration that the sea state may indeed affect the continuous part of the spectrum, and thus the probability distribution may change with the sea state parameters, we have come to the conclusion that it is not possible to deduce the influence of solitons from these exceedance probability plots. Therefore, we have decided not to include the plots in the revised article.

[Figure]

Figure 5: Cumulative exceedance probability of $H/H_s$ in all samples that belong to the category of $0.2 < A_1^S/H_s \leq 0.3$, and two different Ursell number ranges.

**3H** We would like to point out that we do not intend to explain a rogue wave by one soliton alone (as can be seen by comparison of the free-surface elevation ad and the soliton train in Figure 1 in this document). Our hypothesis is that solitons *contribute* to the formation of rogue waves, and we have shown that there is/are always one or several solitons involved when a rogue wave is present in the sample. However, the surface elevation is described not only by solitons, but also by dispersive waves and by the interaction of wave components. This statement is supported by Figure 10 in the preprint, which we therefore would like to keep in the article. Bruehl et al. (2016) have shown that soliton-like waves can form waves that seem to have linear shapes.

**3I** The soliton spectrum alone cannot reveal the formation of rogue waves in general, but solitons may be directly attributed to rogue waves and as such are involved in their presence. We would like to emphasise that it is actually the first time that this has been verified by the nonlinear Fourier analysis of real-world data. Furthermore, as discussed later in the manuscript in relation to Figure 15, certain configurations of the soliton spectrum ($A_2/A_1$) actually do indicate the presence of rogue waves with high probability. The attribution is shown by the method discussed in comment 3G, which we would like to retain in the paper. Furthermore, the size of a soliton is not sufficient to explain the height and the shape of a rogue wave all by itself (see for example Figure 1). As suggested by the reviewer, we will transfer these issues to the discussion of the article to present the line of argument in a straight order.

**3J** The soliton gap was chosen after studying the soliton spectra of many rogue samples. It is not arbitrary, and as far as we know there are no existing alternatives from the literature that would have fit our context. Please also note that many existing signal processing tools heavily rely on the linearity of the transform, and applying them with a nonlinear transform in general is meaningful only in the quasi-linear regime. We do not expect the variance of the soliton spectrum to be a better tool because it would involve solitons that are not associated with the rogue wave. We agree that the conclusion should be that outstanding solitons are not good indicators of rogue waves or large waves near the rogue wave threshold of $H/H_s$ = 2.0, but only for extreme rogue waves. We will avoid the term "predictor" in the text, as the soliton spectrum becomes available only after the recording of the time series and the occurrence of the rogue wave (see also reply to comment 3F).

**3K** The remark is correct, we will adjust the text accordingly, referring to the Forristall distribution (see replies on comments 1B and 2C).

**3L** We confirm that we have not investigated the interaction with oscillatory waves in the context of rogue wave formation and that we are therefore not entitled to make a statement on the exact nature of the interactions. We agree that we should mention this in the conclusions, so as not to raise wrong expectations with the reader. From Figure 1, it is seen that the soliton train alone does not account for the full height of the rogue wave. This only leaves the continuous spectrum for the explanation of the missing height.

**References**

Ablowitz, M. J. & Kodama, Y., 1982. Note on Asymptotic Solutions of the Korteweg-de Vries Equation with Solitons. *Studies in Applied Mathematics,* 66(2), pp. 159-170.

Brühl, M. & Oumeraci, H., 2016. Analysis of long-period cosine-wave dispersion in very shallow water using nonlinear Fourier transform based on KdV equation. *Applied Ocean Research,* Volume 61, pp. 81-91.

Hammack, J. L. & Segur, H., 1974. The Korteweg-de Vries equation and water waves. Part 2. Comparison with experiments. *Journal of Fluid Mechanics,* 65(2), pp. 289-314.

Huntley, D. A., Guza, R. T. & Bowen, A. J., 1977. A universal form for shoreline run-up spectra?. *Journal of Geophysical Research,* Volume 82, pp. 2577-2581.

Osborne, A. R., 1995. The inverse scattering transform: Tools for the nonlinear Fourier analysis and filtering of ocean surface waves. *Chaos Solitons Fractals,* 5(12), pp. 2623-2637.

Osborne, A. R., 2010. *Nonlinear ocean waves and the inverse scattering transform.* Amsterdam: Elsevier.

Osborne, A. R. & Bergamasco, L., 1986. The solitons of Zabusky and Kruskal revisited: Perspective in terms of the periodic spectral transform. *Physica D: Nonlinear Phenomena,* 18(1-3), pp. 26-46.

Osborne, A. R. & Petti, M., 1994. Laboratory-generated, shallow-water surface waves: Analysis using the periodic, inverse scattering transform. *Phys. Fluids,* 6(5), pp. 1727-1744.

Osborne, A. R., Segre, E., Boffetta, G. & Cavaleri, L., 1991. Soliton basis states in shallow-water ocean surface waves. *Physical Review Letters,* Volume 67, pp. 592-595.

Prins, P. J. & Wahls, S., 2019. Soliton Phase Shift Calculation for the Korteweg–De Vries Equation. *IEEE Access,* Volume 7, pp. 122914-122930.

Prins, P. J. & Wahls, S., 2021. An accurate O(N2) floating point algorithm for the Crum transform of the KdV equation. *Communications in Nonlinear Science and Numerical Simulation,* Volume 102, p. 105782.

Schuur, P., 1984. Multisoliton phase shifts in the case of a nonzero reflection coefficient. *Phys. Lett. A,* 102(9), pp. 387-392.

Schuur, P. C., 1986. *Asymptotic Analysis of Soliton Problems: An Inverse Scattering Approach.* 1232 ed. (Lecture Notes in Mathematics): A. Dold and B. Eckmann, Eds. New York, NY, USA: Springer-Verlag.

Sergeeva, A., Pelinovsky, E. & Talipova, T., 2011. Nonlinear random wave field in shallow water: variable Korteweg–de Vries framework. *Natural Hazards and Earth System Sciences,* 11(2), p. 323–330.

Teutsch, I., Weisse, R., Moeller, J. & Krueger, O., 2020. A statistical analysis of rogue waves in the southern North Sea.. *Natural Hazards and Earth System Sciences,* 20(10), p. 2665–2680.

Zabusky, N. J. & Kruskal, M. D., 1965. Interaction of "Solitons" in a Collisionless Plasma and the Recurrence of Initial States. *Physical Review Letters,* Volume 15, pp. 240-243.

---

## Author Comment (AC2)

**Reply to review #2**

We thank Referee #2 for the constructive comments that will help us to clarify and improve several points in our manuscript. In the following, we explain how we plan to address the individual issues raised by the reviewer in the revised manuscript.

**1. On the KdV approximation**

- We fully agree with the reviewer that the assumption that KdV is valid, "cannot be justified on the basis of single-point measurements" (nhess-2022-28-RC2-supplement, p. 1). We would like to point out that we applied vKdV-NLFT as a signal transform, similar to e.g. wavelets or the FFT applied to nonlinear cases. Although we do not know how well the KdV describes the propagation of the measured time series around the measurement site, the KdV does not have to be valid for most of the conclusions of this article, which investigates the results of a signal transform to rogue waves. We do not want to claim that the soliton components in the nonlinear spectrum are physical. We tried to point this out e.g. in the abstract ("Under the hypothesis that the KdV describes the evolution of the sea state around the measurement site well, these results suggest that solitons ...") and the conclusion ("Each measured rogue wave could be associated with at least one soliton in the NLFT spectrum."), but see that this should be pointed out more prominently. We will clarify this in the abstract, the introduction and the conclusion. Our study does not intend to explain the mechanism of rogue wave generation in shallow water. The method should rather be interpreted as a spectral analysis method. We would like to gain insight into the spectral characteristics based on KdV-NLFT at the available measurement site. These spectral characteristics and their differences in samples with and without rogue waves are described in this paper. We would like to point out that in our work, the vKdV-NLFT is applied to a large number of real-world time series for the first time. It is also the first time that certain characteristics of nonlinear spectra could be linked to rogue waves. We thus present a first assessment of the NLFT applied to real measurement data from shallow depths. This is only a first step and future research is needed.
- The reviewer states that "KdV solitons [may usually] be recognized [in time series] by eye" (nhess-2022-28-RC2-supplement, p. 1). We would like to object to this statement and refer to Zabusky and Kruskal (1965), who described the evolution of a sinusoidal-shaped surface elevation, in which solitons eventually form from the background, while not being immediately visible. The observation is reinforced by Brühl & Oumeraci (2016) for the evolution of a long-period cosine wave in very shallow water and in Brühl et al. (2022) for an initially trapezoidal-shaped bore. Here, the solitons that are found by KdV-NLFT, are not immediately visible in the time series, but the surface elevation eventually decomposes into a train of solitons in the far field. While the time series changes with time, the nonlinear spectrum remains invariant. This shows that time series exist, in which KdV solitons are not visible by eye, but may be identified by KdV-NLFT. Another reason for the "invisibility" of the solitons in the time series is that the water surface in the North Sea is not calm before and after the recording of the time series. This means that all existing solitons will continuously interact with the surrounding waves, which makes

them difficult to identify by visual inspection. Figure 1 in Osborne et al. (1991) e.g. demonstrates that solitons do not have to be clearly visible in a real-world measurement.

- The reviewer points out that "the estimated soliton amplitudes are not very large, the solitons do not dominate" (nhess-2022-28-RC2-supplement, p. 1). We have reconstructed a soliton train underlying a time series, by nonlinear superposition of solitons, using the algorithm from Prins & Wahls (2021) (Figure 1 in this document). The time series corresponds to the example in Figure 5 of the preprint. (Note that inverting large soliton spectra is numerically very difficult (Prins & Wahls, 2021). We therefore had to use a shortened time series for the figure.) Figure 1 in this document supports findings by Osborne et al. (1991) that show that the solitons are much lower in amplitude than the maximum waves in the time series. Therefore, we agree with the reviewer, and we have stated so in the conclusion, that in our rogue-wave samples solitons alone cannot be responsible for the formation of the measured rogue waves. The continuous spectrum of the vKdV-NLFT, which actually contains most of the energy in our time series, must account for the remaining parts of the exceptional heights. However, the soliton contribution is not negligible and has the potential to turn a non-rogue wave into a rogue wave. This is concluded from differences in the discrete spectra of samples with and samples without rogue waves.

[Figure]

*Figure 1: a) time series and soliton train, as calculated from the inverted soliton spectrum. b) linear Fourier spectrum. c) continuous spectrum. d) discrete soliton spectrum.*

- We agree with the reviewer to add in the discussion and conclusion that the assumption that waves around our measurement station may be approximated by the KdV equation, cannot be proven based on the available data.
- The reviewer suggests the presence of envelope solitons at varying depths and draws our attention to a study in which "envelope solitons are shown responsible for the wave amplification when the depth increases" (nhess-2022-28-RC2-supplement, p. 1). We agree that varying bathymetry might be an explanation for the enhanced rogue wave occurrence, and will add the corresponding references.
- The reviewer suggests "the oblique interaction of KdV solitons" as an "essentially directional effect" (nhess-2022-28-RC2-supplement, p. 1) as a reason for the enhanced rogue wave occurrence at Norderney. As in the previous comment, we agree with the reviewer that this reason is conceivable, and also, that we cannot assess it in the frames of the KdV equation. All we offer in this article is a signal transformation by NLFT, which

suggests an influence of the presence of shallow-water solitons on rogue wave generation.

- We thank the reviewer for the references on NLS solitons and rogue waves measured in deep water, the accurracy of the NLS equation for strongly nonlinear, the observation of long-lived NLS solitons in the field of strongly nonlinear waves, and KdV soliton content in sinusoids. We will include these in our introduction of the revised article.

**2. Comments regarding the text**

- l. 44: The reviewer points out that the NLS equation is not limited by $kh$ = 1.36. We agree that the formulation in the paper might be misleading and we will clarify this in the revised paper. We will reformulate line 44 in the text to show that the condition is not oriented towards the applicability of the NLS equation, but towards the applicability of the KdV equation.

- l. 83: The reviewer points out that the KdV equation is not the equivalent to the NLS equation, since the former is a wave displacement equation, while the latter is a wave modulation equation. We agree with the reviewer that this formulation is misleading and we will change it accordingly.

- l. 48: We thank the reviewer for the remark that the most unstable perturbations are longitudinal, not oblique. We will replace the word "oblique" by "side-band", as to omit the direction of the perturbation and imply that disturbances arise in the form of side-band modes $\omega(1 \pm \delta)$ to the frequency $\omega$ of the initial wave train.

- l. 53: The reviewer points out that "the BFI parameter was introduced for characterization of irregular wave statistics for the first time in [Onorato et al, (2001)]." (nhess-2022-28-RC2-supplement, p. 2) We agree and we will include this information in the text.

- l. 55: The reviewer points out that breather solutions do not explain the physics of the modulational instability, but describe its dynamics. We agree and we will change the text accordingly.

- l. 61: Thank you, we will insert the correct date of the reference.

- l. 67-73: We thank the reviewer for raising the issue of the shallow water definition. In the context of this study, we have termed the range of applicability of the KdV equation as given by Osborne and Petti (1994), p. 1731, and Osborne (1995), p. 2629, shallow water. We acknowledge that, therefore, our definitions of shallow water ($kh < 1.36$ or $h/L < 0.22$) and deep water ($kh > 1.36$ or $h/L > 0.22$) are different from the definitions of shallow ($h/L < 0.05$), intermediate ($0.05 < h/L < 0.5$) and deep water ($h/L > 0.5$) that are used in the engineering context, and that this may lead to confusion. We will clearly define the terminology for 'shallow' that is used in the paper and repeat the $kh$ value in line 67. Furthermore, we will change the title to include "shallow depths" rather than "shallow water", as suggested by Referee #1, as to avoid confusion of different shallow water terms.

- l. 78: The reviewer points out that rogue waves in variable depths have been discussed in several studies. We will adjust this text accordingly. We thank the reviewer for the additional reference, which we will include.

- l. 97-99: We will refer to the book of Ablowitz and Segur (1981), where in Chapter 1.7c the asymptotic behaviour is discussed for both the continuous and the discrete spectrum. Since the content in this book is somewhat scattered, we also plan to refer to the paper of Ablowitz and Kodama (1982), who correctly analysed the asymptotic behaviour for the first time.

- l. 102: To solve this issue and avoid misunderstandings, we will include the original citation of Pelinovsky et al. (2000): "the "nonlinear" train should include a soliton".

- l. 103, 257: We agree with the reviewer that the interaction of KdV solitons alone does not lead to the formation of the observed rogue waves. We have stated so in lines 104-105. To make this statement more clear, we will reformulate the words "the interaction between one or multiple solitons with oscillatory waves" as "between one or in principle, multiple solitons, with dispersive waves". In line 257, we presume that the nonlinear interaction of solitons with dispersive waves is the probable cause. For clarification, Figure 1 in this document will be added to the revised version. The figure shows an exemplary time series (blue line in the first plot), its linear FFT spectrum (second plot), the nonlinear continuous spectrum (third plot, which will not be analysed further in this article), and the soliton spectrum (last plot). For visualisation of the role of the solitons in the time series, the first plot shows the soliton train (red line) that is obtained by nonlinear superposition of the solitons (considering their interactions) using the algorithm from Prins and Wahls (2021). This example clearly shows that solitons and their interactions are not solely responsible for the generation of the observed rogue wave.

- l. 117: We thank the reviewer for the additional references, which we will include.

- Figure 2: We agree with the reviewer to add a scale for the distance to the map.

- Equation 1: According to the reviewer, the applied shallow-water threshold is related to the BFI and not to the applicability of KdV. We refer to Osborne and Petti (1994), p. 1731 and Osborne (1995), p. 2629 for the shallow-water threshold in KdV and will add the references to the text.

- l. 155: "1" will be removed from the equation.

- Table 1: The caption will be corrected. We agree with the reviewer that due to the recording of waves at discrete sampling points, there is a possibility that the exact crest of a wave is missed. Therefore, it is conceivable that some of the rogue waves termed "height rogue" are actually underestimated and should belong to the category of "extreme rogues". On the other hand, a possible misinterpretation is conservative, thus, all spectral characteristics attributed to extreme rogue samples are assigned correctly. Furthermore, we think that the impact of such effects is small because the sampling frequency is sufficiently large. For the sufficiency of the sampling frequency of our wave buoy, we would like to cite the sampling theorem (Shannon, 1949): if a continuous time signal contains no frequency components higher than W Hz, it may be completely determined by uniform samples taken at the Nyquist rate of $f_s$ = 2W samples per second. A typical (raw) FFT spectrum of a rogue wave time series from our data is shown in the second plot of Figure 1 in this document. It is seen that its components approach zero at approximately 0.5 Hz and have fully decayed at $W$ = 0.64 Hz. The Nyquist rate $f_s$ = 2W = 1.28 Hz is the measurement frequency of the wave buoy. The signal is therefore oversampled by a factor of more than two. We will add a short discussion of this point in the revised paper.

- l. 211: The reviewer points out that the frequency axis of the discrete soliton spectrum indeed has a physical meaning, which is the inverse duration of a soliton. We agree that the formulation may be misleading: although, theoretically, a soliton has an infinitely long duration, since it does not cross the surface, a mathematical definition of the angular frequency can be established (Equation 10). We refer to the soliton solution of the tKdV (see e.g. Equation (12) in Bruehl et al. (2022)), from which the angular frequency may be obtained. We will therefore reformulate the misleading line 211. The reviewer correctly points out that the amplitude and the frequency of a soliton are related. Hence, the frequency gap in the soliton spectrum may be described either in terms of an amplitude ratio or in terms of a frequency ratio. As stated by the reviewer, the relation between the amplitude and the frequency is quadratic. Consequently, a soliton possessing 80% of the amplitude of the

maximum soliton, has a frequency of $\sqrt{0.8} = 0.89 = 89\%$ of the frequency of the maximum soliton. In other words, a reduction in soliton amplitude by 20% corresponds to a reduction in frequency by only 11%. Since the relative differences in the amplitudes are easier to observe in the amplitude-frequency plots, and the soliton amplitude is a more descriptive parameter than its frequency, furthermore, rogue waves are defined in terms of amplitudes, we have selected to use the amplitude ratio instead of the frequency ratio for the definition of outstanding solitons.

- l. 267, 269: The word "right" corresponds to the statistical term used for a distribution drawn over an x-axis. The reviewer is right that this may be misleading, as the distribution is presented as a box plot, with the vertical axis as a reference. We will therefore rephrase the term "to the right" as "towards higher normalised soliton amplitudes".

- l. 276/277 and l. 428/429: As already discussed in the beginning, the term "solitons" was mostly used in the manuscript to refer to components in the discrete spectrum of a time series, which might have been confusing. We will explicitly discuss this issue in the abstract, the introduction, and the conclusion.

We would also like to point out that just like the usual Fourier transform is applied also to signals under nonlinear propagation, we apply the vKdV-NFT to signals that may not propagate according to the KdV. Our results show that the vKdV-NFT, when considered purely as a signal processing tool, leads to interesting new characterizations of certain rogue waves, but also demonstrates limits of this approach (at least in the form used here). How far the soliton components in the nonlinear spectrum are physical is an important question, but that question goes beyond what we can answer with our current data. We nevertheless believe that our work is an important step towards bringing the NFT to real-world data, and hope that it motivates future research in that direction.

- Figure 10: We thank the reviewer for this hint and we will add the dimension to the axis.

- Table 3: The relation mentioned by the reviewer is correct for a single soliton, but the soliton spectrum of more complicated time series changes in more complicated ways with the water depth (Figure 2). We prefer not to put this comment to avoid confusion.

[Figure]

*Figure 2: Soliton spectrum of a rogue wave time series for different water depths assumed in the calculation.*

**References**

Ablowitz, M. J. & Kodama, Y., 1982. Note on Asymptotic Solutions of the Korteweg-de Vries Equation with Solitons. *Studies in Applied Mathematics,* 66(2), pp. 159-170.

Ablowitz, M. J. & Segur, H., 1981. *Solitons and the Inverse Scattering Transform.* Philadelphia: SIAM.

Brühl, M. & Oumeraci, H., 2016. Analysis of long-period cosine-wave dispersion in very shallow water using nonlinear Fourier transform based on KdV equation. *Applied Ocean Research,* Volume 61, pp. 81-91.

Brühl, M. et al., 2022. Comparative analysis of bore propagation over long distances using conventional linear and KdV-based nonlinear Fourier transform. *Wave Motion,* Volume 111, 102905.

Onorato, M., Osborne, A.R., Serio, M., Bertone, S., 2001. Freak Waves in Random Oceanic Sea States. *Physical Review Letters,* Volume 86, pp. 5831-5834.

Osborne, A. R., 1995. The inverse scattering transform: Tools for the nonlinear Fourier analysis and filtering of ocean surface waves. *Chaos Solitons Fractals,* 5(12), pp. 2623-2637.

Osborne, A. R. & Petti, M., 1994. Laboratory-generated, shallow-water surface waves: Analysis using the periodic, inverse scattering transform. *Phys. Fluids,* 6(5), pp. 1727-1744.

Osborne, A. R., Segre, E., Boffetta, G. & Cavaleri, L., 1991. Soliton basis states in shallow-water ocean surface waves. *Physical Review Letters,* Volume 67, pp. 592-595.

Pelinovsky, E., Talipova, T. & Kharif, C., 2000. Nonlinear-dispersive mechanism of the freak wave formation in shallow water. *Physica D: Nonlinear Phenomena,* Volume 147, pp. 83-94.

Prins, P. J. & Wahls, S., 2021. An accurate O(N2) floating point algorithm for the Crum transform of the KdV equation. *Communications in Nonlinear Science and Numerical Simulation,* Volume 102, 105782.

Shannon, C. E., 1949. Communication in the Presence of Noise. *Proceedings of the IRE,* 37(1), pp. 10-21.

Zabusky, N. J. & Kruskal, M. D., 1965. Interaction of "Solitons" in a Collisionless Plasma and the Recurrence of Initial States. *Physical Review Letters,* Volume 15, pp. 240-243.

---

## Referee Report (RR1)

**Review**
on the manuscript "*Contribution of solitons to enhanced rogue wave occurrence in shallow water: a case study in the southern North Sea*" by I. Teutsch, M. Bruhl, R. Weisse, S. Wahls
resubmitted for publication in the **NHESS** journal

The revised version seems to be noticeably improved compared to the original one. However, I still have a significant list of critical remarks. Though I appreciate the work in the whole, sections 1 and 2 are generally poorly written – in the sense that they are obviously authored by a beginner who is not a strong specialist in the theory of nonlinear waves, and in particular the KdV equation. They are barely acceptable. I do not understand this situation, as there are experienced people among the authors.

The introductory part still looks like a collection of unsorted pieces of information which do not form a complete picture. The introduction would benefit if it is shorter but contains clear messages. The selection of references sometimes looks random. For example, I do not understand why the references to Peregrine (1983) are given in lines 120 and 147. It is a good paper, but **it is not at all** about the KdV equation and corresponding IST method. The statement that "*For vanishing boundary conditions, the soliton spectrum completely describes the behaviour of the wave train in the far field*" (Line 264) may be found in any classic book on the IST, not in the recent paper by Prins & Wahls (2019). Eq. (10) is a basic consequence from this statement, therefore the given list of 4 references with equation numbers is absolutely unreasonable. There are other similar examples.

I urge the authors to revise the introductory part of the work, as well as to take into account my comments below, before the article can be recommended for publication.

Line 21. The authors cannot insist that the revealed waves are indeed solitons (i.e., travel preserving their individuality), therefore it seems reasonable to slightly weaken the sentence in the abstract as follows: "*These results suggest that soliton-__like__ and nonlinear processes…*"

Line 35. Please expand the abbreviation "ADCP".

Line 60. The words "*including both nonlinearity and dispersion*" are superfluous and should be removed. This is already said by "*weakly nonlinear narrow-banded approximation*".

Line 75. "*…in terms of linear waves, Stokes waves and breathers*". I believe, this list is not correct. A breather is a coherent wave structure, whereas a Stokes wave is in this sense a free non-linear wave. Therefore I suggest writing as "*in terms of quasi-linear waves and breathers*" or "*in terms of Stokes waves and breathers*".

Line 89. The sentence "*The NLS equation was used as an approximate model of the wave dynamics*" is actually repeats the content of the previous sentence and should be deleted.

Line 99-100. The condition $kh < 1.36$ makes unidirectional waves modulationally stable, what is not sufficient for applicability of the KdV equation. Here and after the authors refer to Osborne & Petti (1994), but these authors discussed the 'cutoff period' for KdV as $kh = 1$ (see their Fig. 3). I did not find a condition of this sort in the second reference Osborne (1995). Bearing in mind that the KdV theory takes into account the two first terms of the Taylor expansion for the dispersion relation ($\tanh(kh) \approx kh -$

1/3 $(kh)^3$), it is obvious that the request of applicability should be at least $(kh)^2 \ll 1$ (assuming that the waves are small enough in amplitude). This comment is also valid for Eq. (1), line 250.

Line 121. It is better to say "*The **regular wave** solutions of the KdV are stable…*"

Line 122. It is better to delete the strange sentence "*This is the mathematical explanation of why rogue waves in shallow water cannot be a result of the modulational instability.*" The modulational instability is absent under the discussed conditions. Therefore the modulational instability cannot be a reason of anything, and a 'mathematical explanation' is not needed.

Line 137. I assume that the sentence "*Costa et al. (2014) found a method to filter soliton trains from measurement data by a linear Fourier transform for the KdV equation with periodic boundary conditions and associating them with wave packets*" is not sufficiently accurate. In Costa et al. (2014) they use the linear Fourier transform to estimate the power law spectrum, which is then shown (using the **nonlinear** method) to be related to solitons. Thus, they use the linear method to see some evidence of hidden solution but not "*to filter soliton trains*".

Fig. 2 caption. What is "NN+m" ?

Line 245. It is not clear from the description, if there were rogue waves satisfying the criteria (5) and (6) simultaneously?

Eqs. (13) and (14). As discussed in the text, these 'different' definitions of Ursell parameter lead to the values which are proportional. Therefore these definitions are essentially the same. The only difference is in the reference (the threshold between soliton- and non-soliton solutions). Besides, what is the need to introduce a new quantity $m$, which is identical to $k^2$?.. These are unnecessary details.

Lines 394-395. The subscripts of amplitudes $A_1$, $A_2$ are given by the regular font, not lower case.

Line 455. The sentence "*Another indication that the soliton spectrum alone is not sufficient to explain the presence of rogue waves is given in Fig. 10, which shows that the shapes of most rogue wave crests are not soliton-like*" is incorrect. It may be concluded from the examples and the discussion provided in the paper, that the revealed solitons have very little in common with the observed extreme wave shapes (see e.g., Fig. 5, top). Therefore they provide no information about rogue wave crests themselves.

---

## Author Response (AR2)

**Reply to Anonymous Referee #1**

We thank Referee #1 for the effort he/she put into revising our manuscript, and the constructive comments that have helped us to a major improvement of the manuscript.

Regarding the additional comment on rogue wave statistics in intermediate water that was suggested by Referee #1:
Following the suggestions of Referee #2, we have strongly revised our Introduction. Its topic is now mainly focused on the analysis of rogue waves using NLFTs. As there was no longer a good location to place the references concerning rogue wave statistics in intermediate water in the Introduction, we have decided to leave the subject of intermediate water rogue wave occurrence frequency in the Discussion, as not to interrupt the new thread of the Introduction.

We also thank the reviewer for pointing out a misleading statement in our previous version. We have adjusted the statement as suggested to "Doeleman (2021) recently showed in tank experiments that the effect of slope is weakened in shallow water." (see line 525 of the revised manuscript, track-changes version).

**Reply to Anonymous Referee #2**

We thank Referee #2 for the additional report, which has helped us re-structure and improve the manuscript.

We have especially put an effort into revising the Introduction section. Following the suggestion of the reviewer, we have re-structured the section to improve the cohesion of the text. The parts regarding nonlinear equations and the NLFT have furthermore been rewritten. We also removed some less relevant parts, and added some sentences to better guide the reader. Furthermore, we have revised our references and chosen classic textbooks where possible. We hope that in this shape, our introductory part appears more convincing to the reviewer and the reader. We finally also improved some formulations and references in the Methods section and split the last paragraph between the Methods and the Results section.

In the following, we address the comments of the referee that concerned specific lines in the manuscript. *The original reviewer comments are listed in italic font.* Our answers to the comments are shown in blue color. The revised lines and mentioned page numbers refer to the new *track-changes version*.

Line 21. *The authors cannot insist that the revealed waves are indeed solitons (i.e., travel preserving their individuality), therefore it seems reasonable to slightly weaken the sentence in the abstract as follows: "These results suggest that soliton-**like** and nonlinear processes…"*
We have changed the sentence in the abstract accordingly (line 21).

Line 35. *Please expand the abbreviation "ADCP".*
Thank you, we have added the full expression to the text (line 35).

Line 60. *The words "including both nonlinearity and dispersion" are superfluous and should be removed. This is already said by "weakly nonlinear narrow-banded approximation".*
We agree with the reviewer and have removed the doubling (lines 64/65).

Line 75. *"…in terms of linear waves, Stokes waves and breathers". I believe, this list is not correct. A breather is a coherent wave structure, whereas a Stokes wave is in this sense a free non-linear wave. Therefore I suggest writing as "in terms of quasi-linear waves and breathers" or "in terms of Stokes waves and breathers".*
The sentence has been removed in the new revision, due to re-formulation of the paragraph (lines 102/103).

Line 89. *The sentence "The NLS equation was used as an approximate model of the wave dynamics" is actually repeats the content of the previous sentence and should be deleted.*
The sentence has been removed in the new revision, due to re-formulation of the paragraph (line 114).

Line 99-100. *The condition kh < 1.36 makes unidirectional waves modulationally stable, what is not sufficient for applicability of the KdV equation. Here and after the authors refer to Osborne & Petti (1994), but these authors discussed the 'cutoff period' for KdV as kh =1 (see their Fig. 3). I did not find a condition of this sort in the second reference Osborne (1995). Bearing in mind that the KdV theory takes into account the two first terms of the Taylor expansion for the dispersion relation (tanh(kh) ≈ kh – 1/3 (kh)3), it is obvious that the request of applicability should be at least (kh)2 ≪ 1 (assuming that the waves are small enough in amplitude). This comment is also valid for Eq. (1), line 250.*
In Osborne & Petti (1994), the KdV cut-off frequency is indeed defined around *kh=1*, but note that this is not considered a strict bound in that paper. They write that "The value $f_{KdV}$ provides a rough definition for a cutoff frequency, far to the right of which, in the spectral domain, KdV evolution cannot occur." (emphasis added for clarity).
In the later paper Osborne (1995), the cut-off frequency $f_{KdV} = 1.36\, c_0/(2\pi h)$ is used in a more strict sense: "Determine if most of the wave energy lies to the left of a KdV 'cutoff frequency,' $f_{KdV} = 1.36\, c_0/(2\pi h)$". Note that this cut-off frequency corresponds to *kh=1.36*, as the following calculation shows. The frequency bound in Osborne (1995) is

$$\frac{1.36\, c_0}{2\pi h} > \frac{1}{T} = f$$

We multiply both sides of the inequality by $\frac{2\pi h}{c_0}$, and arrive at

$$1.36 > \frac{2\pi h}{c_0 T} = \frac{2\pi h}{L} = hk$$

where *L* and *T* are the characteristic spatial and temporal periods and $c_0 = \sqrt{gh}$. The reviewer is correct in that this is the point where the modulational instability disappears.

We updated the description in the paper to reflect that *kh* should be small due to the approximation of the dispersion relation, and that, following Osborne (1995), we used the criterion *kh≤1.36*. The text in the paper has been updated as follows:

"Osborne and Petti (1994) point out that *kh,* with *k* and *h* denoting wave number and water depth, respectively, should not be much larger than one for the KdV equation because of how the dispersion relation is approximated. The threshold kh≤1.36 marks the point at which the modulational instability disappears (ibid.). Following Osborne (1995), we use this threshold to define shallow-water conditions in this work." (lines 150-153).

Line 121. *It is better to say "The regular wave solutions of the KdV are stable…"*
We have added the insertion as suggested (line 153).

Line 122. *It is better to delete the strange sentence "This is the mathematical explanation of why rogue waves in shallow water cannot be a result of the modulational instability." The modulational instability is absent under the discussed conditions. Therefore the modulational instability cannot be a reason of anything, and a 'mathematical explanation' is not needed.*
The reviewer is of course correct in that the modulational instability cannot cause rogue waves if it is absent, but we would like to point this out explicitly for readers that are not familiar with the area. We changed the sentence to "Therefore, the modulational instability cannot contribute to the explanation of rogue-wave occurrence in shallow water." to avoid the "mathematical explanation" part (lines 155/156).

Line 137. *I assume that the sentence "Costa et al. (2014) found a method to filter soliton trains from measurement data by a linear Fourier transform for the KdV equation with periodic boundary conditions and associating them with wave packets" is not sufficiently accurate. In Costa et al. (2014) they use the linear Fourier transform to estimate the power law spectrum, which is then shown (using the nonlinear method) to be related to solitons. Thus, they use the linear method to see some evidence of hidden solution but not "to filter soliton trains".*
We thank the reviewer for this comment. The sentence has been changed to "Costa et al. (2014) used the periodic KdV-NLFT to confirm the soliton content of low-pass filtered time series measured in the Currituck Sound during a storm." (lines 181/182)

Fig. 2 caption. *What is "NN+m" ?*
The height values refer to German "Normalhöhennull", an official vertical datum used in Germany. In the revised version of the manuscript, we have replaced "NN" by the explanation that this represents the standard elevation zero of the German reference height system.

Line 245. *It is not clear from the description, if there were rogue waves satisfying the criteria (5) and (6) simultaneously?*
We did not introduce a specific category to include rogue waves according to both criteria (5) and (6), as this category would have been too small to be statistically significant. Therefore, rogue waves satisfying criterion (6) **can** also satisfy criterion (5), but then they are not treated separately.

Eqs. (13) and (14). *As discussed in the text, these 'different' definitions of Ursell parameter lead to the values which are proportional. Therefore these definitions are essentially the same. The only difference is in the reference (the threshold between soliton- and non-soliton solutions). Besides, what is the need to introduce a new quantity m, which is identical to k2?.. These are unnecessary details.*
We would like to point out that we have added the second definition and further explanation as a reaction to a comment by Referee #1 in the first iteration. However, as not to interrupt

the reading flow, we have now moved the equation and details to a footnote in the updated version of the manuscript. This is also in agreement with the suggestion of Referee #1. (footnote on page 24)

Lines 394-395. *The subscripts of amplitudes A1, A2 are given by the regular font, not lower case.*
We have adjusted the font accordingly.

Line 455. *The sentence "Another indication that the soliton spectrum alone is not sufficient to explain the presence of rogue waves is given in Fig. 10, which shows that **the shapes of most rogue wave crests are not soliton-like**" is incorrect. It may be concluded from the examples and the discussion provided in the paper, that **the revealed solitons have very little in common with the observed extreme wave shapes** (see e.g., Fig. 5, top). Therefore they provide no information about rogue wave crests themselves.*
The reviewer is right. We have removed the sentence in question from the Discussion (lines 503/504).

**References**

Costa, A. et al., 2014. Soliton Turbulence in Shallow Water Ocean Surface Waves. *Phys. Rev. Letters,* 113(108 501).

Doeleman, M. W., 2021. Rogue waves in the Dutch North Sea.

Osborne, A. R., 1995. The inverse scattering transform: Tools for the nonlinear Fourier analysis and filtering of ocean surface waves. *Chaos Solitons Fractals,* 5(12), pp. 2623-2637.

Osborne, A. R. & Petti, M., 1994. Laboratory-generated, shallow-water surface waves: Analysis using the periodic, inverse scattering transform. *Phys. Fluids,* 6(5), pp. 1727-1744.

---

## Author Response (AR3)

**Final Change**

We have re-ordered the formulation of equation (10) and corrected the sentence in lines 255/256, to be correct for the time-like KdV equation.